# TEL: A Thermodynamics-Inspired Layer for Adaptive and Efficient Neural Learning

## Abstract

We introduce the Thermodynamic Equilibrium Layer (TEL), a neural building block that replaces fixed activations with a short, $K$-step energy-guided refinement. TEL performs $K$ discrete gradient steps on a Gibbs-inspired free energy with a learnable step size and an entropy-driven, adaptable temperature estimated from intermediate activations. This yields nonlinearities that are dynamic yet stable, expose useful per-layer diagnostics (temperature and energy trajectories), and run with a fixed, predictable compute budget. Across a broad suite of tasks, TEL matches or exceeds strong baselines, including MLPs, modern implicit/energy-based layers under compute-matched dimensionality, FLOPs, and parameters. Swapping TEL in place of MLP feed forwards in standard different architectural blocks incurs minimal overhead while consistently improving performance. Together, these results position TEL as a scalable, drop-in alternative for constructing adaptable nonlinearities in deep networks.

## 1 Introduction

Multi-layer perceptrons (MLPs) remain a core workhorse of deep learning thanks to their simplicity, universality, and ease of deployment. Classic results show that fully connected feedforward networks can approximate broad classes of nonlinear functions to arbitrary accuracy (Haykin, 1994; Cybenko, 1989; Hornik et al., 1989). In practice, however, the standard composition of linear maps with fixed activation functions (e.g., ReLU, Sigmoid, Tanh) can limit input-dependent adaptivity (Glorot et al., 2011), expressive efficiency (Montúfar et al., 2014; Agostinelli et al., 2014), and robustness under distributional noise and corruption (Hendrycks & Gimpel, 2016). These constraints motivate architectures that provide adaptive, inherently nonlinear transformations while preserving predictable, stable training dynamics.

Recent efforts move toward learnable, input-adaptive nonlinearities. Kolmogorov–Arnold Networks (KANs) (Liu et al., 2024) replace fixed activations with spline functions along edges, improving flexibility and, at times, interpretability, but often at substantial cost: KANs commonly increase parameters and FLOPs by an order of magnitude and can be sensitive to initialization, making them challenging to scale or deploy under tight latency constraints.

We propose the *Thermodynamic Equilibrium Layer* (TEL), a principled alternative to the linear-plus-activation paradigm. TEL models a layer's output as the result of a $K$-step minimization of a Gibbs free-energy functional (Callen & Scott, 1998), akin to a physical system relaxing toward equilibrium. At each refinement step, the update trades off enthalpy minimization against entropy-driven adaptation; the temperature $T$ evolves online from entropy estimates of the activations, providing input-dependent control. This construction yields expressive, adaptive nonlinear transformations while maintaining a fixed, predictable compute budget and exposing useful per-layer diagnostics.

TEL also occupies a distinct point in the design space relative to implicit/equilibrium layers and energy-based approaches. Deep Equilibrium Models (DEQ) solve for a fixed point $z^\star = f_\theta(z^\star, x)$ via root-finding with implicit differentiation, so the computation depends on solver tolerance and can vary across inputs (Bai et al., 2019). Energy-based models (EBM) define an energy landscape and rely on stochastic sampling (e.g., Langevin dynamics) to explore it (LeCun et al., 2006; Grathwohl et al., 2019). In contrast, TEL performs a *bounded*, deterministic $K$-step descent on a Gibbs free-energy objective with an *adaptive temperature*, yielding predictable computation without back-solves or Markov-chain sampling. Empirically, under compute-matched latency and memory bud-

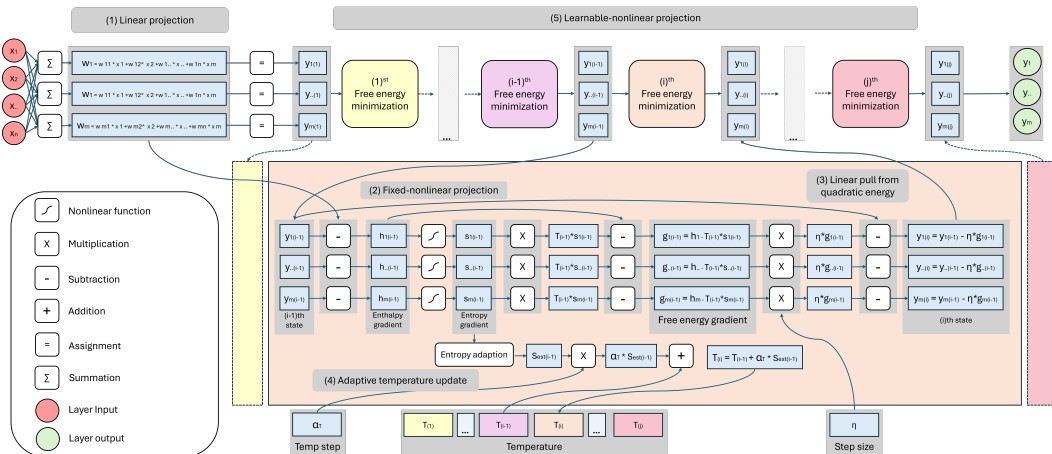

Figure 1: TEL operates in five steps: (1) Linear projection: compute the anchor $Wx$. (2) Fixed nonlinear projection: compute enthalpy and entropy terms. (3) Linear pull: apply the quadratic-energy update toward the anchor. (4) Adaptive temperature update: estimate entropy and update the temperature $T$. (5) Learnable nonlinear projection: apply temperature-scaled nonlinear refinement by iterating for $K$ steps to produce the final output.

gets, TEL competes favorably with DEQ- and EBM-style baselines while using markedly fewer parameters. We evaluate TEL on a broad suite of benchmarks spanning classification, regression, and reconstruction. Across these settings, TEL delivers consistent accuracy gains and favorable accuracy–efficiency trade-offs. Ablations isolate the roles of (i) iterative refinement via $K$-step updates and (ii) entropy-driven temperature adaptation. Our analysis provides conditions ensuring non-expansiveness and bounded gradients under step-size and temperature clipping, and complexity bounds clarifying TEL's parameter and memory profile. We also demonstrate seamless integration of TEL into different standard architectures, supporting its use as a scalable, drop-in building block.

**Contributions.** (1) *Entropy-gradient activations via TEL:* we introduce TEL, which replaces fixed pointwise activations with the gradient of a learned entropy functional, realized as a fixed-$K$ discrete descent on a Gibbs free energy $G_\theta = H_\theta - TS_\theta$. The enthalpy term anchors the layer to the linear projection $Wx$, while the entropy gradient $\nabla_y S_\theta(y)$ serves as an adaptive activation modulated by a data-dependent temperature, yielding predictable compute and per-layer diagnostics (temperature and energy traces) as shown in Figure 1. (2) *Theory & design rules:* conditions for non-expansiveness/contractivity, Lipschitz and gradient-norm bounds, and two-time-scale tracking for the adaptive temperature, leading to simple choices for step sizes, temperature ranges, and clipping. (3) *Empirics:* a three-stage evaluation with shallow building-block analysis, mid-scale backbones, and large-scale benchmarks showing compute-matched gains across different tasks and consistent improvements over MLP/Linear, KANs, and modern implicit/energy-based layers at similar Params/FLOPs.

## 2  THERMODYNAMIC EQUILIBRIUM LAYER (TEL)

TEL is guided by the principle that equilibrium arises from minimizing Gibbs free energy the balance between energy and entropy. We begin by adapting this principle to neural representations (§2.1), then instantiate TEL as a fixed-$K$ iterative refinement with an entropy-adaptive temperature (§2.2), and analyze its expressivity, stability, and efficiency relative to conventional layers (§2.3).

### 2.1  GIBBS FREE ENERGY FOR LAYER COMPUTATION

In thermodynamics, the Gibbs free energy is

$$G \ = \ H \ - \ T\,S, \tag{1}$$

and at fixed temperature and pressure equilibrium corresponds to minimizers of $G$ (Callen & Scott, 1998). We reinterpret this for neural representations by treating the hidden state $y$ as the system state and defining a parameterized free energy

$$G_\theta(y; x, T) = H_\theta(y; x) - T S_\theta(y), \tag{2}$$

where $x$ is the input and $T > 0$ is a (learned and data-dependent) temperature.

We instantiate the enthalpy with a quadratic *anchor* to the linear projection $Wx$,

$$H_\theta(y; x) = \tfrac{1}{2} \|y - Wx\|_2^2, \tag{3}$$

and use an entropy surrogate $S_\theta$ whose gradient defines an activation map $\phi_\theta$, i.e.,

$$\nabla_y S_\theta(y) \triangleq \phi_\theta(y). \tag{4}$$

This yields

$$\nabla_y G_\theta(y; x, T) = (y - Wx) - T \phi_\theta(y). \tag{5}$$

Thus, the temperature $T$ mediates the balance between attraction to the anchor $Wx$ and the nonlinear contribution from $\phi_\theta$. Intuitively, small $T$ contracts $y$ toward the anchor $Wx$ (dominant enthalpy), while larger $T$ emphasizes the nonlinear contribution via $\phi_\theta$ (dominant entropy).

TEL realizes a bounded-compute search for equilibrium by performing $K$ steps of gradient descent on $G_\theta$ with an online *log-temperature* update. To formalize this, we introduce temperature bounds $T_{\min} > 0$, $T_{\max} > T_{\min}$ and their corresponding log-parameters $\tau_{\min} = \log T_{\min}$, $\tau_{\max} = \log T_{\max}$. With $\eta_i > 0$ step sizes and a small dual step $\alpha > 0$,

$$y^{(0)} = Wx, \tag{6}$$

$$T^{(i)} = \exp\big(\mathrm{clip}(\tau^{(i)}, \tau_{\min}, \tau_{\max})\big), \tag{7}$$

$$y^{(i+1)} = y^{(i)} - \eta_i\Big[(y^{(i)} - Wx) - T^{(i)} \phi_\theta\big(y^{(i)}\big)\Big], \qquad i = 0, \dots, K-1, \tag{8}$$

$$\tau^{(i+1)} = \mathrm{clip}\Big(\tau^{(i)} + \alpha\, g_\beta\big(\hat{s}(y^{(i)})\big), \tau_{\min}, \tau_{\max}\Big), \tag{9}$$

where $\hat{s}(y)$ is an entropy estimate computed from activations (analytic or learned) and $g_\beta(z) = \beta_1 z + \beta_0$ is a monotone scaling (optionally applied to an EMA (exponential moving average) of $\hat{s}$ for additional smoothing ) and $\mathrm{clip}(u, a, b)$ denotes elementwise clipping of $u$ to the interval $[a, b]$. Each refinement step first determines the temperature from the clipped log-parameter, then updates the primal state $y$, and finally adjusts $\tau$ through the entropy signal $\hat{s}(y^{(i)})$. The iteration budget $K$ fixes the compute per layer and exposes a practical accuracy–latency knob; full architectural choices for $\phi_\theta$, $\hat{s}$, and $g_\beta$ appear in §2.2.

We choose $\{\eta_i\}$ and $[T_{\min}, T_{\max}]$ to satisfy the non-expansiveness bound in equation 13; in practice, this entails clipping $\eta_i \in (0, 1]$ and selecting $T_{\max}$ such that $T_{\max} L_\phi \leq 1$ to guarantee stable refinement steps.

## 2.2 TEL Architecture

A Thermodynamic Equilibrium Layer (TEL) computes its output via a fixed number $K$ of refinement steps that approximately minimize the free energy in §2.1. Given $x \in \mathbb{R}^{n_{\mathrm{in}}}$, a linear anchor $Wx$ with $W \in \mathbb{R}^{n_{\mathrm{out}} \times n_{\mathrm{in}}}$, and an activation map $\phi_\theta : \mathbb{R}^{n_{\mathrm{out}}} \to \mathbb{R}^{n_{\mathrm{out}}}$ (with $\nabla_y S_\theta(y) \triangleq \phi_\theta(y)$), TEL evolves a hidden state $y$ and returns

$$\Phi_{\mathrm{TEL}}(x; \theta) := y^{(K)}. \tag{10}$$

**Iterative refinement (primal update).** We initialize the hidden state using equation 6, and take $K$ gradient steps on $G_\theta$ using the primal update equation 8. This unrolled refinement defines a depth-$K$ computation in which each step applies the same activation map and temperature rule. Here, $\eta_i > 0$ are learnable step sizes (parameterized in log-space and clipped to a safe range). Parameters of $\phi_\theta$ are *shared across steps*, so TEL's parameter count is essentially independent of $K$ (aside from a few step-size scalars). When $T^{(i)}$ is specified per channel, the product $T^{(i)} \phi_\theta(y)$ is applied elementwise with standard broadcasting semantics.

**Temperature adaptation (log-dual update).** We maintain a log-temperature $\tau$ so that $T = \exp(\tau) > 0$, and update it online from an entropy estimate $\hat{s}(y)$. We reuse the bounds $T_{\min}, T_{\max}$ and their log-parameters $\tau_{\min}, \tau_{\max}$ defined in §2.1; the clipped log-temperature determines the effective temperature via equation 7. The $i$-th refinement step then updates the log-temperature using equation 9, where $\alpha > 0$ is a small dual step and $g_\beta(z) = \beta_1 z + \beta_0$ is a monotone scaling (optionally applied to an EMA of $\hat{s}$ for additional smoothing). TEL supports either a *global* scalar $\tau$ or a *channel-wise* vector $\tau \in \mathbb{R}^{n_{\text{out}}}$ (default: global). At each refinement step we compute $T^{(i)}$ from the clipped log-temperature using equation 7, update $y^{(i+1)}$ via equation 8, and then update $\tau^{(i+1)}$ via equation 9 using $\hat{s}(y^{(i)})$. This dual update allows TEL to adapt its effective nonlinearity on a per-input basis while respecting prescribed temperature bounds.

**Entropy estimator:** We compute $\hat{s}(y)$ from simple activation statistics (e.g., mean/variance, kurtosis) aggregated over batch/spatial axes and (optionally) smoothed with an EMA. TEL supports analytic surrogates (Gaussian/Laplacian/Student-$t$) or a tiny MLP on pooled features; the scale of $\hat{s}$ is absorbed by $g_\beta$, and we detach gradients where noted to avoid degenerate feedback. These choices allow the entropy signal to remain lightweight while still capturing the degree of activation dispersiveness relevant for temperature adjustment.

**Internal free energy vs. training loss:** The free energy $G_\theta(y; x, T)$ is used only to define the layer's internal refinement dynamics. The training objective for TEL networks remains the standard task loss (e.g., cross-entropy for classification, mean-squared error for regression), just as for the baselines; we do not add $G_\theta$ or its entropy term as an explicit regularizer to the global loss. Additionally, the internal step sizes $\eta_i$ govern the $K$-step refinement inside each TEL layer and are distinct from the learning rate of the outer optimizer, which is kept identical across TEL and all baselines. Additional TEL design choices and implementation details are provided in Appendix D.

We initialize $W$ with activation-matched schemes (He/Kaiming for ReLU/SiLU/Swish/GELU; Xavier/Glorot for Tanh) (He et al., 2015; Glorot et al., 2011), set $\eta_i = \eta_0$ initially (shared or per-step), and choose $T_{\min}, T_{\max}$ from a short warm-up. *Shapes.* TEL preserves the tensor shape of $Wx$ and drops into CNN/LSTM/Transformer blocks without reshaping. *Compute.* FLOPs (floating point operations)/latency scale linearly with $K$; training memory is $O(K)$ under standard backprop; rematerialization/checkpointing can reduce this to $O(1)$ at modest extra compute (Chen et al., 2016). TEL also supports optional *early exit* by stopping when the free-energy decrease $\Delta G^{(i)}$ falls below a threshold, enabling adaptive inference cost without architectural branches (Graves, 2016; Teerapittayanon et al., 2016). In practice, these engineering choices make TEL drop-in compatible with standard deep architectures while keeping its overhead modest.

## 2.3 Expressive Properties and Guarantees of TEL

Throughout, let $y^{(0)} = Wx$ and, for $i = 0, \ldots, K-1$, evolve $(y^{(i)}, \tau^{(i)}, T^{(i)})$ via the TEL updates equation 8, equation 7–equation 9. We analyze stability, convergence, and expressivity of the resulting map $\Phi_{\text{TEL}}(x) = y^{(K)}$. This section formalizes the conditions under which TEL behaves as a stable and well-conditioned refinement operator. Formal statements and proofs are collected in Appendix C.

**Assumptions:**

**A1** $\phi_\theta$ is globally $L_\phi$-Lipschitz, i.e., $\|J_{\phi_\theta}(y)\|_2 \le L_\phi$ for all $y$. $J_{\phi_\theta}(y)$ denotes the Jacobian of $\phi_\theta$ at $y$. This ensures controlled nonlinearity across the refinement steps.

**A2** Step sizes satisfy $\eta_i \in (0, \eta_{\max}]$ with $\eta_{\max} < \infty$ (we clip $\log \eta_i$ in practice). Step-size clipping prevents overly aggressive updates that would break non-expansiveness.

**A3** Temperatures are bounded $T^{(i)} \in [T_{\min}, T_{\max}]$ with $0 < T_{\min} \le T_{\max} < \infty$ (enforced by $\tau$-clipping in equation 7–equation 9). This guarantees that TEL's effective gain remains uniformly bounded.

**One–step non-expansiveness and bounded iterates:** Using the primal update in equation 8, we write each refinement step as

$$y^{(i+1)} = \mathcal{F}_i(y^{(i)}), \tag{11}$$

where $\mathcal{F}_i$ is the one-step update map induced by equation 8, and its Jacobian is

$$J_i(y) \;=\; I - \eta_i\big(I - T^{(i)} J_{\phi_\theta}(y)\big). \tag{12}$$

Here $J_{\phi_\theta}(y)$ denotes the Jacobian of $\phi_\theta$ at $y$ and serves as the key object for controlling the Lipschitz behavior of each refinement step.

**Proposition 2.1** (Non-expansive TEL update)**.** *Under **A1–A3**, if*

$$0 < \eta_i \;\leq\; \frac{2}{1 + T_{\max} L_\phi}, \tag{13}$$

*then $\|\mathcal{F}_i(y_1) - \mathcal{F}_i(y_2)\|_2 \leq \|y_1 - y_2\|_2$ for all $y_1, y_2$. If the inequality is strict, $\mathcal{F}_i$ is a contraction (Banach fixed-point theorem applies). The full proof is given in Lemma C.1 in Appendix C.*

**Corollary 2.2** (Boundedness)**.** *Under **A1–A3** and equation 13, the iterates $\{y^{(i)}\}_{i=0}^K$ remain bounded and anchored near $Wx$. In particular, non-expansiveness prohibits divergence even when $K$ is moderately large as a direct consequence of Proposition 2.1 in Appendix C.*

**Convergence with frozen temperature:** Fix $T \in [T_{\min}, T_{\max}]$ and constant $\eta \in (0, \frac{2}{1 + T L_\phi})$. Then the iteration in equation 8 (with $T^{(i)} = T$ and $\eta_i = \eta$) converges linearly to the unique fixed point $y^\star$ solving $y = Wx - T\phi_\theta(y)$, with rate governed by $\max\{|1 - \eta|, \; |1 - \eta(1 - T L_\phi)|\} < 1$ (see Proposition C.4 in Appendix C for details). This provides a baseline convergence guarantee analogous to classical gradient-descent results.

**Two-time-scale tracking with adaptive temperature:** Let the dual step be small relative to the primal, $\alpha \ll \min_i \eta_i$, so that $\tau$ evolves on a slower time-scale. Under **A1–A3**, the coupled dynamics are bounded, and $y^{(i)}$ tracks the instantaneous fixed point $y^\star(T^{(i)})$ with tracking error $O(\alpha)$; see two-time-scale stochastic approximation (Borkar & Borkar, 2008; Konda & Tsitsiklis, 2004) and our formal statement in Proposition C.6 (Appendix C). Intuitively, $y$ nearly equilibrates before $T$ changes appreciably. This separation of time-scales allows TEL to adjust temperature smoothly while retaining near-equilibrium behavior at each iteration.

**Expressivity: recoveries and regimes:** TEL covers several useful regimes:

- **Linear recovery.** If $T^{(i)} \equiv 0$ (or $\phi_\theta \equiv 0$), then $y^{(i)} = Wx$ for all $i$ and TEL reduces to a linear layer.
- **One-step MLP.** With $K = 1$, $y^{(1)} = Wx + \eta_0 T^{(0)} \phi_\theta(Wx)$, recovering a residual MLP-style nonlinearity with data-dependent gain $\eta_0 T^{(0)}$.
- **Implicit-layer limit.** For $K \to \infty$ with fixed $T$, the iteration converges to the solution of $y = Wx - T\phi_\theta(y)$ (if the contraction condition holds), i.e., an implicit/DEQ (Geng & Kolter, 2023) like fixed point obtained by a bounded, controllable solver when $K$ is finite.

These regimes illustrate how TEL interpolates smoothly between classical feedforward layers, residual blocks, and implicit architectures through its temperature and iteration budget.

**Global Lipschitz control:** Under **A1–A3** and equation 13, the end-to-end map $x \mapsto \Phi_{\mathrm{TEL}}(x)$ is globally Lipschitz:

$$L_{\mathrm{TEL}} \;\leq\; \|W\|_2 \prod_{i=0}^{K-1} \|J_i(y^{(i)})\|_2 \;\leq\; \|W\|_2 \prod_{i=0}^{K-1} \max\{|1 - \eta_i|, \; |1 - \eta_i(1 - T_{\max} L_\phi)|\}. \tag{14}$$

Smaller step sizes and tighter temperature bounds decrease $L_{\mathrm{TEL}}$ (improving smoothness/robustness), while larger values increase adaptivity. This provides a direct mechanism for controlling model sensitivity through TEL's design parameters.

**Gradient stability:** By submultiplicativity and equation 13,

$$\left\| \frac{\partial \Phi_{\mathrm{TEL}}(x)}{\partial x} \right\|_2 \;\leq\; \|W\|_2 \prod_{i=0}^{K-1} \max\{|1 - \eta_i|, \; |1 - \eta_i(1 - T_{\max} L_\phi)|\}, \tag{15}$$

Gradients neither explode nor vanish beyond the scaling inherited from $\|W\|_2$ when $(\eta_i, T^{(i)})$ respect the design bounds. In practice, this yields stable training behavior even for moderately large $K$.

**Parameter efficiency and computational complexity:** TEL shares weights across refinement steps: parameters are those of $W$ (and bias), $\phi_\theta$, a few per-step scalars $\{\eta_i\}$, and the log-temperature initialization (global or channel-wise), plus an optional tiny MLP for $\hat{s}(\cdot)$. Thus, the parameter count is essentially independent of $K$. Forward FLOPs consist of one matrix multiply $Wx$ of cost $\Theta(B\,n_{\text{in}}n_{\text{out}})$ plus $K$ elementwise refinements on $y \in \mathbb{R}^{B \times n_{\text{out}}}$ (and a small entropy-estimation overhead), i.e.,

$$\text{FLOPs} \;=\; \Theta(B\,n_{\text{in}}n_{\text{out}}) \;+\; K \cdot \Theta(B\,n_{\text{out}}) \;+\; \text{(estimator overhead)}. \tag{16}$$

Training memory is $O(K)$ under standard backprop; rematerialization/checkpointing can reduce this to $O(1)$ at modest extra compute (Chen et al., 2016). Overall, TEL offers near-constant parameter count with linearly controllable compute, allowing practitioners to tune accuracy–latency trade-offs without architectural redesign. We report latency/throughput/memory trade-offs in §3 and provide full FLOP and memory derivations in Appendix E.

In the next section, we investigate how these stability and efficiency guarantees translate into empirical performance under compute-matched budgets and ablations of TEL's design choices.

## 3 EXPERIMENTS

We structure our empirical evaluation in three stages that progressively increase architectural complexity. This staged design lets us isolate what TEL contributes as a building block, then test its robustness as we move from toy settings to standard backbones and finally to large-scale benchmarks. Because TEL introduces an internal $K$-step refinement, its behavior can depend on depth, width, and surrounding modules, so disentangling these regimes is essential. Comprehensive training details are presented in Appendix G, and all supplementary results per-dataset curves, full tables, and extended ablations are provided in Appendix H.

**Stage I: Shallow building-block analysis.** We start with the minimal setting: a single hidden layer where the standard MLP block is replaced by TEL. This removes confounders such as depth, skip connections, normalization, and attention, and allows us to (i) measure the intrinsic benefit of iterative refinement, (ii) compare TEL directly to other equivalent building blocks under strictly matched width, parameter count, and FLOPs, and (iii) Identify a robust default configuration for TEL through ablations. These experiments show that TEL improves accuracy/error and reduces variance even without depth, indicating that its gains come from the refinement dynamics themselves.

**Stage II: Mid-scale backbones.** We then insert TEL into lightweight, widely used CNN/LSTM/Transformer architectures. This stage probes whether TEL is a practical drop-in replacement for feedforward blocks in standard networks and compares two deployment patterns: *TEL-head* (replace only the first MLP block) and *TEL-full* (replace all such blocks). These models are still small enough to keep effects interpretable but rich enough to include convolution, recurrence, and attention.

**Stage III: Large-scale benchmarks.** Finally, we test TEL in larger benchmarks with more complex tasks. This stage evaluates whether TEL's thermodynamic refinement remains stable and beneficial at scale, and whether the *TEL-head* pattern continues to be preferable in deep residual and attention-based models.

Across all three stages, TEL layers are configured to satisfy the design constraints of §2.3 (step-size clipping, temperature bounds, and non-expansive refinement), so performance differences can be attributed to TEL's mechanism rather than to ad-hoc tuning.

### 3.1 STAGE I: SHALLOW BUILDING-BLOCK ANALYSIS

Stage I evaluates TEL in the most controlled setting possible: We evaluate the Thermodynamic Equilibrium Layer (TEL) across classification, regression, and reconstruction benchmarks MNIST, Fashion-MNIST, CIFAR-10/100, STL10 (classification); standard UCI datasets such as Diabetes, Energy, Concrete, Wine, and California Housing (regression); and synthetic manifolds including 1D sinusoids, 2D moons/spirals, and 3D swiss roll/spheres for autoencoding (reconstruction). Baselines include Linear, MLP (Linear+ReLU), KAN (Liu et al., 2024), EBM (Du & Mordatch, 2019), and DEQ (Geng & Kolter, 2023). Stage I evaluates TEL in the most controlled setting, a single hidden

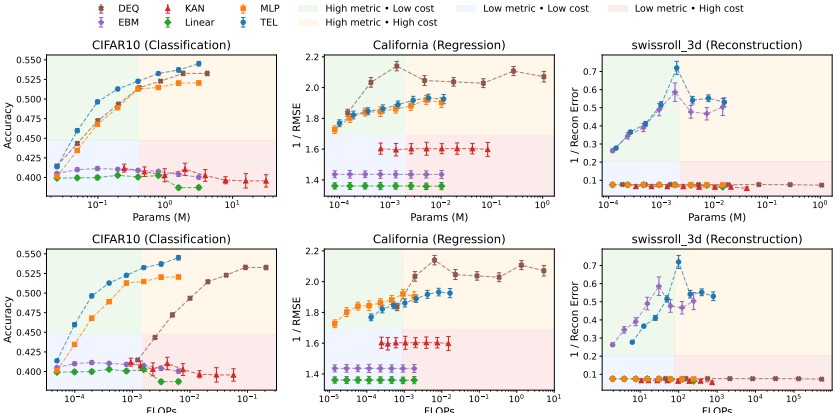

Figure 2: Average performance ($\pm$ std) over 20 runs across 5 different random seeds, evaluated on three representative datasets, one each for classification, regression, and reconstruction using six building-block models: Linear, MLP, KAN, EBM, DEQ, and TEL. Results are plotted against parameter count and FLOPs for hidden embedding sizes ranging from 8 to 1028.

layer where TEL directly replaces the (Linear+ReLU) block, ensuring matched width, parameter count, and FLOPs across all models and removing confounding architectural factors. This provides a clean test of TEL's intrinsic properties, including refinement, temperature adaptation, and stability.

**Performance at Matched Width, Params, and FLOPs** Fig. 2 reports representative results on one dataset per task CIFAR-10 (classification), California (regression), and the Swishroll 3D (reconstruction). Across these examples, TEL consistently lies on or dominates the Pareto frontier in terms of accuracy (or RMSE / reconstruction error), parameter count, and FLOPs.

Matched width (8–1028): TEL outperforms all baselines at the same width in **35/40** classification comparisons, **31/40** regression comparisons, **32/40** reconstruction comparisons. TEL's improvements appear even at very small widths (16–64), where the benefits of iterative refinement and temperature adaptation are most pronounced. This demonstrates that TEL is intrinsically more expressive than standard one-step nonlinearities. For results across all 15 datasets, see Appendix H.1.

**Matched parameter count:** TEL dominates the "low-params, high-performance" quadrant. Because TEL reuses parameters across $K$ steps, increasing $K$ improves accuracy without increasing parameters moving TEL *vertically* in Pareto plots. KAN and DEQ must increase parameter count or depth to match TEL's frontier. Full results across the 15 datasets are provided in Appendix H.2.

**Matched FLOPs:** TEL more frequently occupies the "low-FLOPs, high-performance" region than any other method. DEQ and EBM close the gap only at substantially higher FLOPs due to solver or sampling overhead. TEL's cost grows deterministically with $K$, making it predictable and tunable. Results for all 15 datasets are shown in Appendix H.2.

**Stability Across Seeds** TEL consistently shows smaller variance often 2–3× lower than KAN and EBM, and 1.5–2× lower than DEQ across all tasks and widths. This confirms that TEL's refinement dynamics are stable to initialization and stochasticity. TEL's coefficient-of-variation remains the lowest across nearly the entire width sweep.

**Ablations** Iteration budget ($K$): Fig. 3 shows clear diminishing returns for $K > 5$ across all tasks. The accuracy/error curve exhibits a knee at $K \in [3, 5]$. FLOPs grow linearly, parameters remain fixed. This matches TEL's theoretical contraction-based refinement (§2.3).

**Temperature adaptation:** Fig. 4 reports results on the three representative datasets, where adaptive temperature (global or channel-wise) consistently outperforms fixed or learned-static temperature.

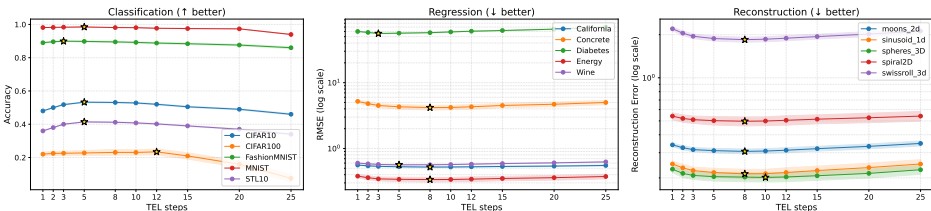

Figure 3: Effect of iteration budget $K$. Mean performance vs. $K$ over 15 datasets and hidden dimension 256; shaded bands are 95% CIs. FLOPs scale linearly with $K$; parameters are constant. The knee appears at $K \approx [3\text{–}8]$.

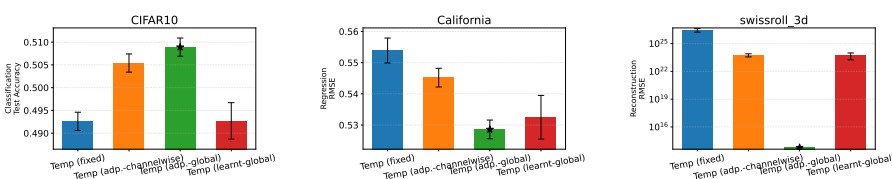

Figure 4: Performance at hidden dimension 256 and Temperature schemes. Fixed $T$ vs. adaptive $T_t$ with (i) a Gaussian estimator and (ii) a 2-layer MLP estimator, each in global and channel-wise variants. Adaptive $T_t$ consistently outperforms fixed $T$.

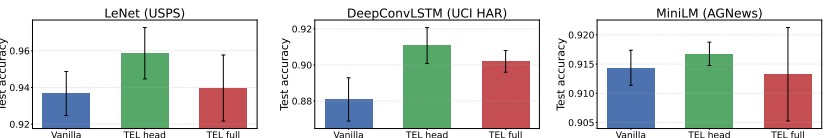

Figure 5: **TEL as a drop-in replacement in common backbones.** Mean $\pm$std test accuracy over 20 runs (5 random seeds) when swapping linear+ReLU blocks with TEL in three settings: LeNet, DeepConvLSTM, and MiniLM. We compare the original (Vanilla), replacing only the first FFN with TEL (TEL-head), and replacing all FFNs (TEL-full).

Global adaptive temperature provides the strongest gains with minimal parameter cost. Full results across all 15 datasets are provided in Appendix H.4.

From the shallow-layer analysis, four conclusions emerge consistently: (i) TEL provides sizable gains in accuracy/RMSE/reconstruction even in the absence of depth; (ii) TEL is more parameter- and FLOP-efficient than all competing building blocks; (iii) TEL demonstrates markedly lower variance and stronger stability across seeds; (iv) the most reliable operating regime is $K \in [3, 5]$ with adaptive global temperature. This configuration is used throughout the deeper evaluations in §3.2 and §3.3.

## 3.2 STAGE II: MID-SCALE BACKBONES

We next evaluate TEL inside lightweight CNN, LSTM, and Transformer backbones, using LeNet (CNN) (LeCun et al., 2002), DeepConvLSTM (LSTM) (Ordóñez & Roggen, 2016), and MiniLM (Transformer) (Wang et al., 2020) on the USPS image dataset (Van der Maaten, 2009), the UCI HAR time-series dataset (Nayak et al., 2022), and the AGNews natural-language dataset (Tang et al., 2019), respectively. This stage examines whether the trends from Stage I persist once TEL is embedded in deeper, modality-diverse networks.

As provided in Figure 5, replacing only the *first* feedforward block with TEL (TEL-head) yields consistent accuracy gains of +0.8–1.5% in CNNs, +0.9–1.3% in LSTMs, and +0.6–1.0% in Transformers, all with negligible parameter overhead. These improvements mirror the shallow-layer find-

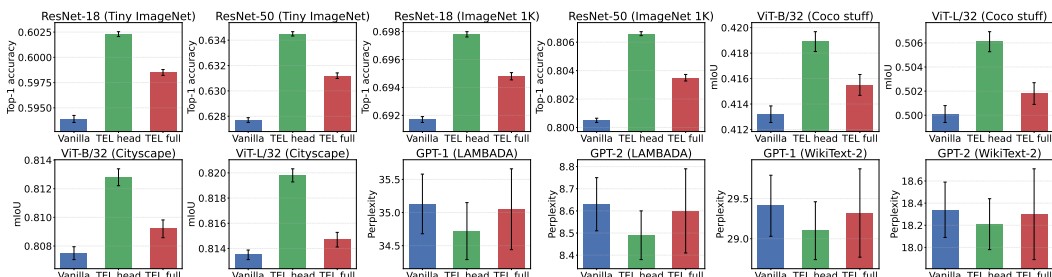

Figure 6: Mean ± std. performance across all high-capacity backbones and datasets: *ResNet-18/50 on Tiny-ImageNet and ImageNet-1K*, *ViT-B/L on COCO-Stuff and Cityscapes*, and *GPT-1/2 on LAMBADA and WikiText-2*. Three configurations per model: *vanilla*, *TEL-head*, and *TEL-full*. TEL-head consistently provides the largest improvements, while TEL-full remains close behind. All models are trained with matched FLOPs and parameter budgets (Appendix G).

ings of Stage I and indicate that TEL strengthens early representations across modalities. Replacing all feedforward blocks (TEL-full) also improves over the vanilla architectures, but typically underperforms TEL-head, reinforcing the emerging pattern that TEL is most impactful as an early-stage feature refiner.

Across all architectures, TEL variants exhibit low run-to-run variance (std $< 0.5\%$), continuing the stability advantages first observed in Stage I.

TEL introduces only negligible compute overhead: parameter increases remain below $0.2\%$ for CNNs, $0.05\%$ for LSTMs, and $0.5\%$ for Transformers, with FLOP changes $< 0.3\%$ in all cases. TEL-head increases latency modestly, while TEL-full incurs larger but still manageable slowdowns. These small costs accompany consistent +0.6–1.5% accuracy gains, making TEL-head the best accuracy efficiency tradeoff. Detailed parameter, FLOP, and latency measurements for all models appear in Appendix H.5.

Overall, TEL acts as a practical, architecture-agnostic, and parameter-efficient drop-in replacement for feedforward blocks. TEL-head is consistently the strongest configuration and is therefore adopted as the default in Stage III.

## 3.3 STAGE III: LARGE-SCALE BENCHMARKS

We evaluate TEL in high-capacity architectures ResNet-18/50 (He et al., 2016), ViT-B/L (Dosovitskiy, 2020), and GPT-1/2 (Radford et al., 2018; 2019) across ImageNet-1K (Deng et al., 2009), Tiny-ImageNet (Le & Yang, 2015), COCO-Stuff (Caesar et al., 2018), Cityscapes (Cordts et al., 2016), LAMBADA (Paperno et al., 2016), and WikiText-2 (Merity et al., 2016), covering both large-scale vision benchmarks and complex language reasoning tasks. This stage tests whether the refinement behavior observed in Stages I–II persists in deep residual networks, attention-based models, and large transformers operating on substantially more challenging datasets and tasks.

As shown in Figure 6, TEL-head yields consistent accuracy gains across all architectures. For classification, TEL-head improves ResNet-18 by $+1.4\%$ on Tiny-ImageNet and $+0.9\%$ on ImageNet-1K, and improves ResNet-50 by $+1.1\%$ and $+0.8\%$. For segmentation, it increases ViT-B/L mIoU by $+0.7\%$–$1.4\%$. For language modeling, TEL-head reduces GPT-1/2 perplexity by $0.7\%$–$1.6\%$. These patterns mirror earlier stages: TEL most strongly impacts early-layer representations.

TEL-full also improves performance but with consistently smaller gains and higher compute. For example, improvements in ResNets and ViTs fall to $+0.4\%$–$0.8\%$, and GPT models show only marginal perplexity changes. This confirms that TEL's refinement is most beneficial in early layers.

TEL-head adds only modest overhead $< 1\%$ more FLOPs, and moderate latency increases. TEL-full, by contrast, introduces substantially larger slowdowns. Comprehensive runtime, parameter, and FLOP analyses for all architectures are provided in Appendix H.6.

In summary, TEL scales reliably to ResNets, ViTs, and GPT-style transformers. TEL-head offers a strong accuracy efficiency tradeoff with robust improvements across all modalities, while TEL-full remains functional but yields diminishing returns with depth.

### 3.4 INTERPRETABILITY & DIAGNOSTICS

TEL's refinement dynamics expose internal thermodynamic quantities—enthalpy and entropy gradients, temperature updates, and free-energy trajectories—that are both theoretically grounded and empirically diagnostic. Unlike feedforward MLPs or implicit layers, which offer no reliable internal observables, TEL provides physically interpretable signals tightly correlated with sample difficulty, uncertainty, and the stability of the refinement process.

Across shallow, mid-scale, and large-scale architectures, four diagnostic families appear with striking consistency: (i) the enthalpy–entropy gradient ratio $\rho^{(i)}$, which separates anchor- dominated from entropy-driven refinement and exhibits a clear difficulty ordering; (ii) the gradient alignment $\kappa^{(i)}$, whose dips reflect disagreement between enthalpy and entropy updates and highlight geometrically challenging or atypical samples; (iii) the temperature trajectory $T^{(i)}$ and its mean $\overline{T}$, which rise more strongly for harder examples and track epistemic uncertainty; and (iv) the free-energy evolution $\Delta G^{(i)}$, which decreases smoothly under TEL's stability constraints and serves as a simple convergence or early-exit criterion. Comprehensive visualizations and per-example analyses appear in Appendix H.7.

### 3.5 LIMITATIONS

While TEL shows promising performance–efficiency trade-offs and stable refinement dynamics, several limitations constrain the scope of our claims.

**Stacking depth:** Because a single TEL block already performs multiple iterative refinements in parallel, stacking many TEL layers would introduce several nested optimization processes, increasing training complexity and often leading to instability. Consequently, even in deep architectures, we primarily use TEL in its *single-layer* form (TEL-head) within each block. Despite this restriction, TEL-head still provides meaningful improvements, much like diffusion-style refinement layers or functional primitives such as KAN, where most of the gains arise from a single functional layer rather than deep stacking. Thus, TEL acts as a high-capacity substitute for the MLP sub-layer rather than a primitive designed for multi-layer stacking. Developing mechanisms that could support stable, deeper TEL stacks, such as cross-block residual pathways, propagating intermediate equilibrium states, or sharing temperature priors across layers, remains valuable future work.

**Diagnostics are not fully interpretable:** Although TEL exposes diagnostic signals such as temperature, entropy, and free energy, these quantities provide heuristic guidance rather than strict interpretability or guarantees. Unlike symbolic or explicitly structured methods (e.g., spline-based KANs), TEL's diagnostics are informative but not yet actionable. Closing this gap between useful indicators and fully interpretable or verifiable behaviors remains an important research direction.

## 4 CONCLUSION

We introduced the Thermodynamic Equilibrium Layer (TEL), a drop-in adaptive nonlinearity that replaces fixed activations with a short $K$-step free-energy refinement. TEL provides input-dependent nonlinear behavior with predictable computation, comes with simple design rules ensuring non-expansiveness and stable gradients, and integrates seamlessly into standard architectures. Across all three evaluation stages, shallow building-block analysis, mid-scale backbones, and large-scale models TEL-head consistently delivers performance improvements under matched width and tightly controlled Params/FLOPs. TEL's gains persist even when used as a single layer inside deep architectures, offering a strong accuracy efficiency trade-off with negligible parameter overhead and modest latency increase. The thermodynamic formulation also provides useful diagnostic signals, which correlate with sample difficulty, uncertainty, and refinement dynamics.

Future work includes designing stackable TEL primitives and improving the interpretability of TEL's diagnostic signals.

ETHICS STATEMENT

All authors have read and will adhere to the ICLR Code of Ethics. This work introduces a generic neural layer (TEL) and evaluates it on standard, publicly available datasets (vision, text, and tabular); no human subjects were recruited, and no personally identifiable information was collected. We comply with dataset and software licenses and disclose no conflicts of interest or external sponsorship that could unduly influence results. Potential risks include amplification of dataset biases and disparate behavior from dynamic inference (early exit) across subpopulations. To mitigate this, we report mean±std over multiple seeds, recommend subgroup analyses where labels permit, and expose diagnostics ($\overline{T}, \Delta G, \rho, \kappa$) to audit uncertainty, convergence, and shift. TEL's compute scales with the refinement budget $K$; we use early stopping and encourage energy/carbon tracking when scaling. No new datasets are released, and no sensitive domains (e.g., surveillance, biometric identification) are targeted.

REPRODUCIBILITY STATEMENT

We aim for full reproducibility. The main text specifies the model and update rules (Secs. 2.1–2.2), theoretical assumptions and guarantees (Sec. 2.3), and the experimental protocol (Sec. 3). The appendix includes related work, additional proofs, design choices for enthalpy/entropy/temperature, complexity analysis with early exit, and a step-by-step algorithm. An anonymous repository with code, configs, and scripts is included in the appendix: it provides exact scripts, dataset preprocessing, FLOPs/parameter counting, and figure scripts; training uses fixed $K$ with early exit disabled, and all hyperparameters are specified in config files.

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

## APPENDIX

## A  CODE AVAILABILITY

An anonymous repository with full source code, configs, and scripts to reproduce all results is provided here: **[https://anonymous.4open.science/r/TEL-04C8/README.md]**. The repo includes a PyTorch implementation of TEL, dataset download/preprocessing scripts, one-line run commands for width sweeps and Pareto plots, and figure notebooks. The link will be de-anonymized upon acceptance.

## B  RELATED WORK

**Adaptive nonlinearities and learned activations.** A long line of work replaces fixed pointwise nonlinearities with learned or input-adaptive variants to improve expressivity and optimization. Early approaches include leaky/parametric rectifiers and ELU (Maas et al., 2013; He et al., 2015; Clevert et al., 2015), as well as smoother gates like GELU and Swish (Hendrycks & Gimpel, 2016; Ramachandran et al., 2017). Gating-based feedforward layers (e.g., GLU/SwiGLU) and mixture-of-experts route inputs through input-dependent sub-functions (Dauphin et al., 2017; Shazeer, 2020;

Shazeer et al., 2017). Dynamic activations further condition parameters on the input (e.g., Dynamic ReLU) (Chen et al., 2020). Kolmogorov–Arnold Networks (KANs) learn spline functions on edges, offering flexibility and some interpretability at a considerable parameter/FLOP cost (Liu et al., 2024). TEL differs from these one-shot transformations by performing a short, bounded sequence of refinement steps on a free-energy $G_\theta$, with an adaptive temperature $T$ that yields input-dependent behavior under a predictable $K$-step compute budget and exposes per-layer diagnostics.

**Implicit/equilibrium layers and differentiable optimization.** Implicit deep learning replaces explicit stacks with the solution of an equilibrium or optimization problem. Deep Equilibrium Models (DEQ) solve for a fixed point $z^\star = f_\theta(z^\star, x)$ via root finding with implicit differentiation; the computation depends on solver tolerance and can vary across inputs (Bai et al., 2019; 2020). Differentiable optimization layers embed QP/convex program solvers inside networks (Amos & Kolter, 2017; Agrawal et al., 2019), and continuous-depth models evolve hidden states via ODE solvers (Chen et al., 2018). TEL contrasts with these by using a deterministic, *fixed $K$-step* descent on a Gibbs-inspired objective $G_\theta$, avoiding back-solves while retaining solution-driven semantics and exposing temperature/energy trajectories.

**Energy-based models and thermodynamic perspectives.** Energy-based models (EBMs) define unnormalized densities $p_\theta(y \mid x) \propto \exp\{-E_\theta(y; x)\}$ and are typically trained with MCMC or score-based dynamics (LeCun et al., 2006; Grathwohl et al., 2019; Du & Mordatch, 2019; Welling & Teh, 2011; Song & Ermon, 2019). They can be powerful but incur stochastic sampling cost and mixing concerns. TEL shares the energy perspective, minimizing a free-energy functional inspired by statistical physics (Callen & Scott, 1998) but uses short, deterministic refinement steps with an entropy-driven temperature; no Markov-chain sampling or negative-phase estimation is required. Related thermodynamic/equilibrium ideas include Equilibrium Propagation and modern Hopfield networks (Scellier & Bengio, 2017; Ramsauer et al., 2020); TEL leverages a free-energy view for *layer-level* computation rather than network-level training or associative memory.

**Stability, Lipschitz control, and diagnostics.** Constraining networks to be approximately non-expansive improves robustness and stabilizes training (Cisse et al., 2017; Miyato et al., 2018; Tsuzuku et al., 2018; Gouk et al., 2021). TEL provides simple design rules, step-size, and temperature clipping that ensure non-expansiveness ($\mathrm{Lip} \leq 1$) and bounded gradients at the layer level (proved in our analysis). Beyond accuracy, TEL exposes temperature and energy traces that function as diagnostics during training and inference, complementing work connecting flatter minima to generalization (Hochreiter & Schmidhuber, 1997; Keskar et al., 2016; Li et al., 2018).

TEL combines strengths of adaptive activations (input dependence), implicit layers (solution-driven semantics), and energy-based views (principled objectives) while maintaining a *fixed iteration budget* and providing per-layer diagnostics. Empirically, we compare against KANs (Liu et al., 2024), DEQ-style implicit layers (Bai et al., 2019), and EBM-inspired baselines (Grathwohl et al., 2019) under compute-matched budgets, highlighting TEL's accuracy–efficiency Pareto advantages.

## C  ADDITIONAL PROOFS

**Standing assumptions and notation.** We adopt **A1–A3** from the main text (global Lipschitz activation, clipped step sizes, and bounded temperatures). The TEL refinement map is exactly the update rule in equation 8 with temperature defined by equation 7 and log-temperature update equation 9; we do not restate these equations here.

When useful, we also invoke:

**A0 (elementwise $\phi_\theta$).** $\phi_\theta$ acts coordinatewise and is differentiable a.e., with $0 \leq \phi'_\theta(z) \leq L_\phi$ for all $z$. Then for any $y_1, y_2$ there exists a diagonal $D(y_1, y_2)$ with spectrum in $[0, L_\phi]$ such that $\phi_\theta(y_1) - \phi_\theta(y_2) = D(y_1, y_2)(y_1 - y_2)$. *Remark:* **A0** $\Rightarrow$ **A1**; we use **A0** only to sharpen constants.

**Lemma C.1** (One-step Lipschitz bound). *Let $\mathcal{F}_i$ denote the one-step refinement operator induced by the TEL update equation 8. Under **A0–A3**, for all $y_1, y_2$,*

$$\|\mathcal{F}_i(y_1) - \mathcal{F}_i(y_2)\|_2 \leq \rho_i \|y_1 - y_2\|_2, \qquad \rho_i \triangleq \max\{|1 - \eta_i|, \; |1 - \eta_i(1 - T^{(i)} L_\phi)|\}. \quad (17)$$

*Proof.* This follows by applying **A0** to the TEL update equation 8 and collecting terms exactly as done in the main-text discussion preceding Proposition 2.1. Explicitly, letting $\Delta = y_1 - y_2$,

$$\mathcal{F}_i(y_1) - \mathcal{F}_i(y_2) = \big[(1 - \eta_i)I + \eta_i T^{(i)}D(y_1, y_2)\big]\Delta,$$

whose operator norm equals $\max_{d \in [0, L_\phi]} |1 - \eta_i(1 - T^{(i)}d)|$. $\qquad\square$

**Proposition C.2** (Non-expansiveness; contraction under the design bound). *Under **A1–A3**, the design rule equation 13 in the main text implies the bound*

$$0 < \eta_i \ \le\ \frac{2}{1 + T_{\max}L_\phi},$$

*which ensures $\rho_i \le 1$ and hence that $\mathcal{F}_i$ is non-expansive. If the inequality is strict, $\mathcal{F}_i$ is a contraction.*

*Proof.* The worst case is achieved at $d = L_\phi$ and $T^{(i)} = T_{\max}$. Substituting this into the expression for $\rho_i$ from Lemma C.1 yields the claim. $\qquad\square$

**Corollary C.3** (Bounded refinements). *Under **A1–A3** and equation 13, the sequence $\{y^{(i)}\}_{i=0}^K$ generated by the TEL refinement remains bounded and anchored near $Wx$.*

**Proposition C.4** (Linear convergence for frozen $T$). *Assume **A1–A3**. Fix $T \in [T_{\min}, T_{\max}]$ and constant $\eta \in (0, \, 2/(1 + TL_\phi))$. Then the frozen-$T$ iteration*

$$y^{(i+1)} = \mathcal{F}_T(y^{(i)})$$

*converges linearly to the unique fixed point $y^\star$ solving $y = Wx - T\phi_\theta(y)$, with rate $\rho(\eta, T) = \max\{|1 - \eta|, |1 - \eta(1 - TL_\phi)|\} < 1$. This is the main-text result referenced in §2.3.*

**Lemma C.5** (Lipschitz dependence of $y^\star$ on $T$). *Under **A1–A3** with $TL_\phi < 1$, the fixed point $y^\star(T)$ satisfies*

$$\|y^\star(T_1) - y^\star(T_2)\| \ \le\ \frac{\sup_{y \in \mathcal{Y}} \|\phi_\theta(y)\|}{1 - T_{\max}L_\phi} \, |T_1 - T_2|.$$

*Proof.* The argument follows directly from the implicit function theorem applied to the fixed-point equation $F(y, T) = y - Wx + T\phi_\theta(y) = 0$, using the contraction condition established in the main text. $\qquad\square$

**Proposition C.6** (Two-time-scale tracking with adaptive $T$). *Assume **A1–A3**, equation 13, and that $|\hat{s}(y)| \le S_{\max}$ by clipping. Let $T^{(i)} = \exp(\tau^{(i)})$ with $\tau^{(i)}$ updated by equation 9. If $\alpha \ll \min_i \eta_i$, then the tracking error $e^{(i)} := y^{(i)} - y^\star(T^{(i)})$ obeys*

$$\|e^{(i+1)}\| \ \le\ \bar{\rho}\,\|e^{(i)}\| + C\,\alpha, \qquad \sup_i \|e^{(i)}\| \ \le\ \frac{C}{1 - \bar{\rho}}\,\alpha,$$

*where $\bar{\rho} < 1$ and $C$ depends on $S_{\max}$, $L_g$, $e^{\tau_{\max}}$, and Lemma C.5.*

*Proof.* This follows by decomposing the error into the frozen-$T$ contraction term (from Proposition C.4) plus the drift in $y^\star(T)$, bounded using Lemma C.5 and the clipped change in $\tau^{(i)}$ specified by equation 9. $\qquad\square$

**Lemma C.7** (Gradient norm bound). *Under **A1–A3** and equation 13,*

$$\left\|\frac{\partial \Phi_{\mathrm{TEL}}(x)}{\partial x}\right\|_2 \ \le\ \|W\|_2 \prod_{i=0}^{K-1} \max\big\{|1 - \eta_i|, \ |1 - \eta_i(1 - T^{(i)}L_\phi)|\big\}.$$

*Proof.* Immediate from the chain rule applied to the Jacobians of the per-step update equation 8, together with the bound in Lemma C.1. $\qquad\square$

**Remarks.** (i) The elementwise condition **A0** is needed only for Lemma C.1; all other results require only **A1–A3**. (ii) The design rule (clipped $\eta_i$, bounded $T_{\max}$ so that $T_{\max}L_\phi \le 1$) guarantees equation 13. (iii) Replacing $\hat{s}$ by an EMA affects only constants in Proposition C.6 and not the qualitative behavior of the two-time-scale argument.

# D  Design Choices

## D.1  Enthalpy

TEL's enthalpy term is the quadratic anchor already used implicitly in the primal refinement rule equation 8. The gradient $\nabla_y H_\theta(y; x) = y - Wx$ defines a 1-Lipschitz "harmonic well" centered at $Wx$, and the corresponding equilibrium condition

$$0 = (y^\star - Wx) - T\,\phi_\theta(y^\star)$$

is precisely the stationarity relation associated with the TEL refinement map.

Because equation 8 uses the free-energy gradient defined in §2.1, the global Lipschitz bound and non-expansiveness properties follow directly from Lemma C.1 and Proposition 2.1 in Appendix C. In particular, the design rule equation 13 guarantees stable refinement without needing to restate any additional equations here.

More structured anchors—such as anisotropic quadratics, robust (Huber / $\ell_1$) penalties, Bregman divergences, or graph-regularized forms can be incorporated by replacing the unit Lipschitz constant of the quadratic anchor with a generic $L_H$ in Proposition 2.1; the stability condition remains of the form equation 13 with 1 replaced by $L_H$.

Empirically, the simple quadratic anchor offers the best trade-off between stability, interpretability, and ease of tuning, and is therefore used by default in all experiments. Alternative anchors are only warranted when one wishes to encode explicit geometry (e.g., anisotropy, sparsity, graph structure) and is willing to adjust $L_H$ and the temperature range accordingly.

## D.2  Entropy

TEL models entropy via the activation force $\phi_\theta(y)$ appearing in the refinement update equation 8; equivalently, $\nabla_y S_\theta(y) = \phi_\theta(y)$ is already built into the free-energy gradient.

The stability analysis in Lemma C.1 and Proposition 2.1 depends only on the global Lipschitz constant $L_\phi$ of $\phi_\theta$, so any activation satisfying Assumption **A1** inherits the same non-expansiveness guarantees. No additional analytic constraints are required beyond this Lipschitz bound.

Different nonlinearities instantiate different entropy geometries inside the same refinement rule equation 8: ReLU variants provide sparse, piecewise-linear forces; tanh/sigmoid provide smooth, saturating forces; Swish/SiLU/GELU provide smooth, non-piecewise forces with good conditioning; ELU-family activations introduce asymmetric shaping on negative values; and learned activations allow data-adaptive entropy geometry. Empirically, Swish/SiLU and PReLU perform best; we adopt Swish/SiLU as the default entropy gradient due to its strong performance and zero additional parameter cost, and use learned or PReLU-type activations only in targeted ablations.

## D.3  Temperature: Dual Variable and Adaptive Control

Temperature $T$ is updated via the log-temperature rule equation 7–equation 9 defined in the main TEL architecture and referenced throughout Appendix C, we summarize their conceptual role.

The primal TEL evolution is given by the refinement update equation 8, while the dual update of $\tau = \log T$ is equation 9. Together, these form the two–time–scale system analyzed in Proposition C.6, which shows that, under bounded temperatures and the design rule equation 13, the iterates $y^{(i)}$ track the instantaneous fixed point $y^\star(T^{(i)})$ up to an $\mathcal{O}(\alpha)$ error. Intuitively, $y$ nearly equilibrates at the current temperature before $T$ changes appreciably.

In practice, we treat $T$ as a control variable that balances the anchor and entropy forces in $G_\theta$. A small dual step $\alpha$, clipping of $\tau$ (and hence $T$), and an entropy estimate entering through equation 9 suffice to keep the temperature trajectory smooth. All constants governing stability and tracking error follow directly from the analysis in Appendix C.

Practical guidance is simple: choose $\eta_i$ to satisfy equation 13, use a comparatively small dual step $\alpha$, clamp $T$ via equation 9, and optionally detach gradients through the entropy estimator to avoid feedback loops. These settings were used in all experiments and were sufficient to keep TEL's refinement stable across architectures and datasets.

## D.4 ENTROPY ESTIMATION

Entropy estimation appears in TEL only through the log-temperature update equation 9, which depends on a scalar (or channel-wise) proxy $\hat{s}(y)$. Since the refinement rule equation 8 is fixed, the estimator affects only the dual dynamics and not the form of the primal update itself.

We compute $\hat{s}(y)$ from the entropy force $z = \phi_\theta(y)$ by pooling over batch and spatial/sequence axes:

$$\mu_c = \text{mean}(z_c), \qquad \sigma_c^2 = \text{var}(z_c) + \varepsilon.$$

**Analytic estimators.** Gaussian, Laplacian, and Student-$t$ estimators are used in their standard closed forms and only rescale the input to equation 9. A robust Gaussian variant with $\tilde{\sigma} = \kappa \, \text{MAD}(z)$ improves stability for small or noisy batches and is used as our default analytic choice.

**Learned estimator.** A tiny MLP processes summary moments and outputs a global or channel-wise entropy proxy. Because the TEL update equations remain those of equation 8 and equation 9, the only requirement on the learned estimator is to respect the Lipschitz assumptions needed for the temperature dynamics; in practice, we enforce this via a simple slope and weight-norm constraints and, when helpful, by stopping gradients through $\hat{s}(y)$.

Overall, Gaussian or robust-Gaussian estimators provide the best stability–cost trade-off in most settings, Student-$t$ is helpful in the presence of heavy tails, and MLP-based estimators slightly improve cross-dataset calibration at negligible extra compute. In all cases, the estimator's granularity (global vs. channel-wise) is chosen to match the corresponding temperature variant in equation 9.

## E COMPLEXITY ANALYSIS OF TEL

We detail parameters, FLOPs/latency, and memory of a TEL layer, align notation with the main text, and describe *early exit* for inference.

**Parameter count.** For input width $n_{\text{in}}$ and output width $n_{\text{out}}$:

- Linear map $W \in \mathbb{R}^{n_{\text{out}} \times n_{\text{in}}}$ (+bias): $n_{\text{in}} n_{\text{out}}$ (+$n_{\text{out}}$).
- Step sizes: $\{\eta_i\}_{i=0}^{K-1}$ typically $K$ scalars (optionally per-step vectors if desired).
- Log-temperature init(s): $\tau^{(0)}$ one scalar (global) or $n_{\text{out}}$ (channel-wise).
- Entropy estimator: analytic (Gaussian/Laplace/$t$) adds *no* params; a tiny MLP on pooled moments adds $O(h \cdot d)$ where $d$ is the number of pooled statistics (e.g., mean/var/kurtosis per channel) and $h \in [16, 64]$.

Hence, with the common (global-$T$, analytic-estimator) choice:

$$\#\text{params} = n_{\text{in}} n_{\text{out}} + O(n_{\text{out}}) + O(K) \tag{18}$$

i.e., essentially the same as a linear layer (and far below spline-based KANs that scale like $O(n_{\text{in}} n_{\text{out}} G)$ for grid size $G$).

**FLOPs and latency (forward).** Let $B$ be batch size and let $y \in \mathbb{R}^{B \times n_{\text{out}}}$ denote the hidden state after the anchor $Wx$.

$$\text{Anchor: } Wx \text{ costs } \Theta(B \, n_{\text{in}} n_{\text{out}}) \quad \text{(once).}$$

$$\text{Per refinement step: } \phi_\theta(y), \; g_H = y - Wx, \; T^{(i)} \odot \phi_\theta(y), \; \text{axpy} \Rightarrow \Theta(B \, n_{\text{out}}).$$

$$\text{Estimator (optional): pool moments + tiny MLP} \Rightarrow \tilde{O}(B \, n_{\text{out}}) \text{ (negligible).}$$

With a *fixed* budget $K$:

$$\text{FLOPs}_{\text{forward}} = \Theta(B \, n_{\text{in}} n_{\text{out}}) + K \cdot \Theta(B \, n_{\text{out}}) + \text{(estimator overhead)} \tag{19}$$

**Training memory and backward cost.** Unrolling $K$ refinements yields $O(K)$ activation memory under standard backprop; gradient checkpointing/rematerialization reduces this to $O(1)$ with a modest extra forward pass per checkpoint. Backward FLOPs scale like forward FLOPs (within a small constant).

**Early exit at inference (adaptive $K$).** TEL supports a *fixed* maximum budget $K$ and a per-example effective budget $K_{\text{eff}}(x) \leq K$ decided on-the-fly by a cheap stopping rule. We use either:

**(i) Gradient-norm rule:** $\quad \|g_G^{(i)}\| \;=\; \big\| (y^{(i)} - Wx) - T^{(i)}\phi_\theta(y^{(i)}) \big\| \;\leq\; \varepsilon_{\text{grad}}$ for $m$ consecutive steps,

**(ii) Energy-decrease rule:** $\quad |\Delta G^{(i)}| \;=\; \big| G_\theta(y^{(i+1)}; x, T^{(i)}) - G_\theta(y^{(i)}; x, T^{(i)}) \big| \;\leq\; \varepsilon_G$ for $m$ steps,

with patience $m \in \{1, 2\}$ to avoid premature exits. Rule (i) avoids computing $S_\theta$ explicitly; rule (ii) provides a literal free-energy criterion when $S_\theta$ is available.

Let $q_j$ be the probability of exiting exactly at step $j$ ($1 \leq j < K$) and $q_{\geq K}$ the probability of using the full budget. Then

$$\mathbb{E}[K_{\text{eff}}] = \sum_{j=1}^{K-1} j\, q_j + K\, q_{\geq K}, \qquad \text{FLOPS}_{\text{infer}} = \Theta(B\, n_{\text{in}} n_{\text{out}}) + \mathbb{E}[K_{\text{eff}}] \cdot \Theta(B\, n_{\text{out}}) \quad (20)$$

In practice, we apply early exit *only at inference*; training uses a fixed $K$ for stable gradients.

## F  ALGORITHM

We have provided a simplified TEL layer visualization in Figure 7 accompanied by the Algorithm 1 for better understanding of its implementation.

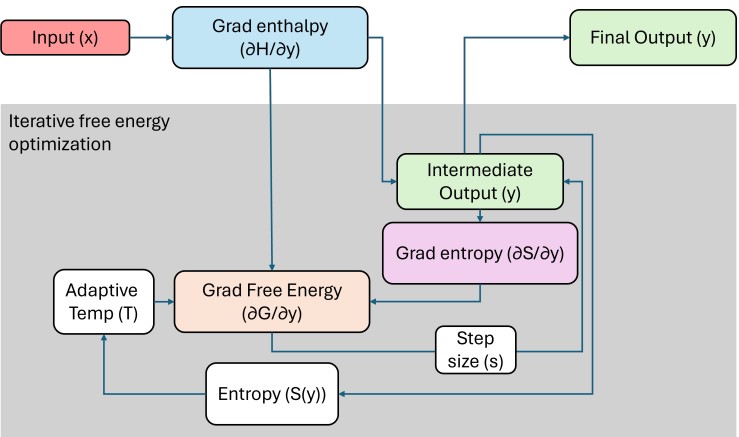

Figure 7: TEL at a glance simplified. Each TEL layer begins with a linear projection $Wx$ (the *enthalpy anchor*) and refines a hidden state $y$ via $K$ iterations that minimize a Gibbs free-energy objective. At iteration $i$, the update balances the enthalpy gradient $(y - Wx)$ against the entropy gradient $\phi(y)$, scaled by an *adaptive temperature* $T$. The temperature evolves online from entropy estimates of the activations, yielding input-dependent adaptivity. The result is a bounded, nonlinear transformation with a fixed and predictable iteration budget.

(i) $\eta_i$ and $\tau$ can be global scalars or channel-wise vectors; broadcasting is elementwise. (ii) To ensure the non-expansive regime, pick $T_{\max}$ and clip $\eta_i$ so that $0 < \eta_i \leq 2/(1 + T_{\max}L_\phi)$. (iii) For inference, freezing $\tau$ avoids distributional drift; if adaptation at test-time is desired, reduce $\alpha$ and keep tight $\tau$ bounds. (iv) Early exit changes *latency* but not *parameters*; it is disabled during training to keep gradients well-defined.

**Computational cost.** Each TEL iteration requires:

1. One matrix-vector multiplication $Wx$ ($O(n_{\text{in}} n_{\text{out}})$).

2. One application of activation $\phi$ ($O(n_{\text{out}})$).

3. Entropy estimation:
   - Analytic: mean/variance computation ($O(Bn_{\text{out}}$ for batch size $B$).

---

**Algorithm 1** TEL forward with adaptive temperature and early exit.

---

**Require:** $x \in \mathbb{R}^{n_{\text{in}}}$; $W \in \mathbb{R}^{n_{\text{out}} \times n_{\text{in}}}$; activation $\phi_\theta$; steps $\{\eta_i\}_{i=0}^{K-1}$; log-temp init $\tau^{(0)}$ (global or channel-wise); bounds $\tau_{\min}, \tau_{\max}$; dual step $\alpha$; estimator $\widehat{s}(\cdot)$; scaler $g_\beta$; flags TRAIN $\in$ {True, False}, EARLY_EXIT $\in$ {True, False}; thresholds $\varepsilon_{\text{grad}}, \varepsilon_G$; patience $m$.

1: $y^{(0)} \leftarrow Wx$             ▷ Enthalpy anchor
2: $\tau^{(0)} \leftarrow \text{clip}(\tau^{(0)}, \tau_{\min}, \tau_{\max})$
3: streak $\leftarrow 0$
4: **for** $i = 0$ **to** $K - 1$ **do**
5:      $T^{(i)} \leftarrow \exp\big(\text{clip}(\tau^{(i)}, \tau_{\min}, \tau_{\max})\big)$
6:      $g_H \leftarrow y^{(i)} - Wx$             ▷ Enthalpy gradient
7:      $g_S \leftarrow \phi_\theta\big(y^{(i)}\big)$             ▷ Entropy gradient
8:      $g_G \leftarrow g_H - T^{(i)} \odot g_S$             ▷ Free-energy gradient
9:      $y^{(i+1)} \leftarrow y^{(i)} - \eta_i \odot g_G$             ▷ Primal update
10:      **if** TRAIN **then**             ▷ Dual update during training
11:          $s_i \leftarrow \widehat{s}\big(g_S\big)$ **(EMA/clamp as needed)**
12:          $\tau^{(i+1)} \leftarrow \text{clip}\big(\tau^{(i)} + \alpha\, g_\beta(s_i), \tau_{\min}, \tau_{\max}\big)$
13:      **else**
14:          $\tau^{(i+1)} \leftarrow \tau^{(i)}$             ▷ Freeze $\tau$ by default at inference
15:      **end if**
16:      **if** EARLY_EXIT **and** $\neg$ TRAIN **then**
17:          **Option A (default):** $\kappa \leftarrow \|g_G\|_2$; **Option B:** $\Delta G \leftarrow G_\theta(y^{(i+1)}; x, T^{(i)}) - G_\theta(y^{(i)}; x, T^{(i)})$
18:          **if** (Option A: $\kappa \leq \varepsilon_{\text{grad}}$) **or** (Option B: $|\Delta G| \leq \varepsilon_G$) **then**
19:              streak $\leftarrow$ streak $+ 1$
20:              **if** streak $\geq m$ **then break**             ▷ Early exit at $K_{\text{eff}} = i + 1$
21:          **end if**
22:          **else**
23:              streak $\leftarrow 0$
24:          **end if**
25:      **end if**
26: **end for**
27: **return** $y^{(\text{last})}$             ▷ Output at $K_{\text{eff}}$ (inference) or $K$ (training)

---

- MLP: additional $O(Bd_h n_{\text{out}})$ where $d_h$ is the hidden size (constant or small).

Repeating for $K$ refinement steps, the total forward cost is

$$\text{FLOPs} \approx K \cdot \big(O(n_{\text{in}} n_{\text{out}}) + O(B n_{\text{out}})\big). \tag{21}$$

Backward cost is at most a constant factor larger, as all operations are differentiable. Runtime is predictable given the fixed $K$.

**Memory usage.** Memory is dominated by storing activations $y^{(i)}$ for $i = 0, \ldots, K$, which requires $O(KBn_{\text{out}})$. Entropy statistics add $O(Bn_{\text{out}})$ per step. Total memory, therefore, scales as

$$O(KBn_{\text{out}} + n_{\text{in}} n_{\text{out}}), \tag{22}$$

which is comparable to deep MLPs and significantly lighter than spline-based KANs (which must store grid evaluations).

**Comparison summary.**

- **MLP layer:** $O(n_{\text{in}} n_{\text{out}})$ params, $O(Bn_{\text{in}} n_{\text{out}})$ FLOPs.
- **KAN layer:** $O(n_{\text{in}} n_{\text{out}} G)$ params, $O(Bn_{\text{in}} n_{\text{out}} G)$ FLOPs.
- **TEL layer:** $O(n_{\text{in}} n_{\text{out}} + K)$ params, $O(KBn_{\text{in}} n_{\text{out}})$ FLOPs, predictable by iteration budget $K$.

TEL achieves adaptive nonlinear transformations with complexity close to an MLP, and with substantially fewer parameters than KANs. The key trade-off is a factor $K$ in compute, which is controllable and modest in practice (e.g., $K = 5$–$10$ suffices).

## G  TRAINING AND EXPERIMENTAL SETUP

This appendix provides full experimental details for all models and datasets, complementing the protocol description in §3. Unless stated otherwise, all models—including TEL and every baseline—are trained under *identical* optimization settings, hyperparameter search grids, early-stopping criteria, and data preprocessing pipelines. This ensures that performance differences arise from the choice of layer (TEL vs. baseline) rather than differences in training procedure.

### G.1  SHARED OPTIMIZATION AND FAIRNESS PROTOCOL

**Optimizer and gradient handling.    Optimizer.** We use the AdamW optimizer with default

$$(\beta_1, \beta_2) = (0.9, 0.999), \qquad \varepsilon = 10^{-8}. \tag{23}$$

Unless otherwise stated, the weight decay is fixed to $10^{-2}$ for all methods. Gradients are clipped to have Euclidean norm at most

$$\|g\|_2 \leq 1.0 \tag{24}$$

before each optimizer step.

**Learning-rate schedule.**    Unless noted otherwise, we use cosine decay with warmup:

$$\text{lr}(t) = \lambda_0 \cdot \begin{cases} t/T_{\text{warmup}}, & t < T_{\text{warmup}}, \\ \frac{1}{2}\left(1 + \cos\left(\pi \frac{t - T_{\text{warmup}}}{T_{\text{max}} - T_{\text{warmup}}}\right)\right), & t \geq T_{\text{warmup}}, \end{cases} \tag{25}$$

with $T_{\text{warmup}} = 5$ epochs and $T_{\text{max}}$ the maximum epoch budget (see below).

**Training budget, early stopping, and model selection.**    All models are trained for up to $T_{\text{max}} = 1000$ epochs with early stopping on the validation metric:

- classification: validation accuracy,
- regression: validation RMSE,
- reconstruction: validation reconstruction error,
- segmentation: validation mIoU,
- language modeling: validation perplexity.

We use patience 15 epochs: training stops if the validation metric does not improve for 15 consecutive epochs. All performance numbers in the main text and appendix are reported using the checkpoint with the best validation value.

**Batch size and hardware.**    We use a batch size of 512 for all tasks and methods, trained on NVIDIA GPUs:

- Stages I–II: a single RTX 6000 Ada (48GB).
- Stage III: training on 4×A100 GPUs for efficiency, but all reported latency and throughput are measured on a single RTX 6000 Ada.

**Hyperparameter grids and fairness.**    For each dataset family (vision classification, tabular regression, synthetic reconstruction, sequence classification, segmentation, and language modeling), we sweep the following shared hyperparameter grids for all baselines (Linear, MLP, EBM-style refinement, DEQ, and KAN) and TEL:

- **Learning rate:**
$$\{1 \times 10^{-4},\ 3 \times 10^{-4},\ 1 \times 10^{-3},\ 3 \times 10^{-3}\}. \tag{26}$$
- **Dropout:**
$$\{0.0,\ 0.1,\ 0.2\} \tag{27}$$
for fully connected and Transformer-style models.

- **Weight decay:**

$$\{0, \ 10^{-2}\}. \tag{28}$$

For each model class and dataset, the hyperparameter configuration with the best validation performance (averaged over seeds) is selected and used to report test metrics.

**Random seeds and repeated runs.** Unless otherwise stated, all reported numbers are computed as a mean and standard deviation over multiple independent runs. For Stage I and II benchmarks, we use

$$5 \text{ random seeds } \times \ 4 \text{ independent runs } = \ 20 \text{ runs per dataset}, \tag{29}$$

and report the mean and standard deviation over these 20 runs. For all other benchmarks, we use 5 random seeds and report mean $\pm$ standard deviation over these 5 runs.

## G.2   TEL-Specific Hyperparameters

TEL introduces a small set of additional internal parameters that control the K-step refinement. These do not affect the global training loss or outer optimizer; they only change the internal layer dynamics.

**Step-size parameters.** Each TEL layer maintains $K$ step sizes $\eta_i$ parameterized in log-space,

$$\eta_i = \exp(\tilde{\eta}_i), \tag{30}$$

and clipped to the range

$$\eta_i \in [\eta_{\min}, \eta_{\max}] = [10^{-4}, 1.0]. \tag{31}$$

Unless otherwise stated, $\eta_i$ are independent learnable scalars (shared across channels but not across steps).

**Temperature bounds.** We set a minimum and maximum temperature

$$T_{\min} = 0.05, \qquad T_{\max} \text{ such that } T_{\max} L_\phi \leq 1, \tag{32}$$

following the non-expansiveness condition in §2.3. We parameterize temperature via $\tau = \log T$ and clip

$$\tau \in [\tau_{\min}, \tau_{\max}] = [\log T_{\min}, \log T_{\max}]. \tag{33}$$

We use a global scalar $\tau$ per TEL layer by default; channel-wise $\tau$ is used in some ablations and yields similar trends.

**Dual step for temperature.** The dual learning rate $\alpha$ controls how fast the log-temperature evolves:

$$\alpha \in \{5 \times 10^{-3}, \ 10^{-2}, \ 2 \times 10^{-2}\}, \tag{34}$$

with the best validation choice used per dataset. We always ensure $\alpha \ll \min_i \eta_i$ so that the temperature evolves on a slower time-scale than the primal refinement (§2.3).

**Entropy estimator.** Unless otherwise noted, we use a simple Gaussian-entropy surrogate computed from the batch mean and variance of the activations. Concretely, for a channel-wise activation vector $y$ with empirical variance $\hat{\sigma}^2$, we define

$$\hat{s}(y) \propto \frac{1}{2} \log(\hat{\sigma}^2 + \varepsilon), \tag{35}$$

and optionally smooth this estimate with an exponential moving average with half-life 5. The scale of the entropy estimate is normalized through the affine map

$$g_\beta(z) = \beta_1 z + \beta_0 \tag{36}$$

with $\beta_1 \in \{0.5, 1.0, 2.0\}$ and $\beta_0 = 0$. For large-scale ViT and GPT experiments, we additionally consider a tiny 2-layer MLP on pooled statistics as an entropy estimator; this adds $< 0.01\%$ parameters.

**Iteration budget.** TEL refines its hidden state with a fixed budget of $K$ refinement steps. Except where noted, we use $K = 5$ in all main comparisons. The ablation in Fig. 3 sweeps

$$K \in \{1, 2, 3, 5, 8, 10, 12, 15, 20, 25\} \tag{37}$$

on all 15 Stage I datasets (5 classification, 5 regression, 5 reconstruction), with identical width and outer optimization across all $K$.

### G.3 DATASETS

We evaluate TEL in three stages. Stage I uses 15 shallow tasks (5 classification, 5 regression, 5 reconstruction) to analyze TEL as a building block. Stage II inserts TEL into mid-scale backbones on 3 classification tasks. Stage III evaluates TEL at scale on 2 image classification tasks, 2 semantic segmentation tasks, and 2 autoregressive language modeling tasks. All datasets use their standard splits and evaluation protocols unless noted.

#### G.3.1 STAGE I: SHALLOW BUILDING-BLOCK ANALYSIS (15 DATASETS)

**Classification (5 datasets).** These match Fig. 2 in the main text.

- **MNIST.** 60k training and 10k test images of size $28 \times 28$ (grayscale), 10 classes (LeCun et al., 2002). Pixel values are normalized to $[0, 1]$. No augmentation.
- **Fashion-MNIST.** Same structure as MNIST (60k/10k, $28 \times 28$, 10 classes) but with clothing categories (Xiao et al., 2017). Normalization identical to MNIST; no augmentation.
- **CIFAR-10.** 50k training and 10k test RGB images of size $32 \times 32$, 10 classes (Krizhevsky et al., 2009). Preprocessing: per-channel mean/variance normalization. Augmentation: random crop with 4-pixel padding, horizontal flip with probability $0.5$.
- **CIFAR-100.** Same image format as CIFAR-10 but 100 classes (Krizhevsky et al., 2009). Preprocessing and augmentation identical to CIFAR-10.
- **STL-10.** 5k labeled train images, 8k test images, and 100k unlabeled images of size $96 \times 96$ (Coates et al., 2011). We downsample to $64 \times 64$, apply per-channel normalization, and use random resized crop + horizontal flip for augmentation.

**Regression (5 datasets).** We use five tabular regression tasks following the standard UCI-style protocol (Asuncion et al., 2007):

- **California** (housing).
- **Concrete** (compressive strength).
- **Diabetes** (Physiological variables).
- **Energy** (energy efficiency).
- **Wine** (wine quality).

For each dataset, we standardize all features to zero mean and unit variance. We follow a 20-split protocol: each split uses 80% of the data for training/validation (further split 80/20 internally) and 20% for testing; we report mean $\pm$ std of test RMSE across the 20 splits.

**Reconstruction (5 datasets).** We use five synthetic manifolds for autoencoder reconstruction, following common benchmarks for nonlinear manifold learning:

- **sinusoid 1D** (1D sinusoidal curve).
- **moons 2D** (two interleaving half circles).
- **spiral2D** (2D spiral).
- **spheres 3D** (points on one or multiple spheres).
- **swissroll 3D** (3D Swiss roll).

Each dataset is normalized to zero mean and unit variance per coordinate. We train autoencoders with a 2D bottleneck and use mean-squared error reconstruction loss; performance is reported as 1/Recon. Error as in Fig. 2.

### G.3.2 STAGE II: MID-SCALE BACKBONES (3 CLASSIFICATION TASKS)

**USPS.** Grayscale $16 \times 16$ digit images, 10 classes (Van der Maaten, 2009). We follow the standard USPS train/test split.

**UCI HAR.** Human Activity Recognition dataset with multivariate time series from smartphone accelerometer and gyroscope (Nayak et al., 2022). We use the standard train/test split and preprocessing with channel-wise normalization and fixed-length windows.

**AGNews.** News topic classification with 4 classes (Tang et al., 2019). We use the standard train/test split.

### G.3.3 STAGE III: LARGE-SCALE BACKBONES (2+2+2 TASKS)

**Image classification (2 datasets).**

- **Tiny-ImageNet.** 100k training and 10k validation images across 200 classes (Le & Yang, 2015). Images are resized to $64 \times 64$; augmentation includes random resized crop and horizontal flip. We evaluate ResNet-18 and ResNet-50 (He et al., 2016) with and without TEL.
- **ImageNet-1K.** 1.28M training and 50k validation images across 1000 classes (Deng et al., 2009). We use the standard $224 \times 224$ pipeline: resize $\rightarrow$ random crop $\rightarrow$ horizontal flip. We evaluate ResNet-18 and ResNet-50 (He et al., 2016) in vanilla, TEL-head, and TEL-full configurations.

**Semantic segmentation (2 datasets).**

- **COCO-Stuff.** 164k images with 171 semantic segmentation classes (Caesar et al., 2018). We use ViT-B/32 and ViT-L/32 backbones (Dosovitskiy, 2020) with standard segmentation heads. Augmentation: resize, scale jitter (0.5–2.0), random crop, horizontal flip. Metric: mean IoU (mIoU) on the validation split.
- **Cityscapes.** 5k high-resolution urban street scenes with 19 classes (Cordts et al., 2016). We use ViT-B/32 and ViT-L/32 (Dosovitskiy, 2020) with the same augmentation protocol as for COCO-Stuff. Metric: mIoU.

**Autoregressive language modeling (2 datasets).**

- **LAMBADA.** A long-range word prediction benchmark (Paperno et al., 2016). We use GPT-1 and GPT-2 style decoder-only Transformers (Radford et al., 2018; 2019), with byte-level BPE tokenization. Metric: perplexity.
- **WikiText-2.** Word-level language modeling dataset (Merity et al., 2016). We use the same GPT-1/GPT-2 backbones (Radford et al., 2018; 2019) as for LAMBADA, with identical tokenizer and vocabulary. Metric: perplexity.

### G.4 ARCHITECTURES AND TEL INSERTION POINTS

TEL is inserted differently depending on model family. We summarize here how TEL replaces or augments standard MLP/FFN blocks in each architecture, grouped by the three experimental stages. In all cases, *TEL-head* means "replace the **first** MLP/FFN block in the backbone", and *TEL-full* means "replace *all* such blocks".

### G.4.1 STAGE I: SHALLOW MLP AND AUTOENCODER

**Shallow MLP (classification and regression).** The base model is a single-hidden-layer MLP:

$$x \in \mathbb{R}^{d_{\text{in}}} \xrightarrow{W_1} h \in \mathbb{R}^d \xrightarrow{\sigma} z \xrightarrow{W_2} \hat{y}, \tag{38}$$

where $d$ is the hidden width (varied in $\{8, 16, 32, 64, 128, 256, 512, 1024\}$), and $\sigma$ is the nonlinearity. Baselines use $\sigma = \text{ReLU}$ or the corresponding DEQ/EBM/KAN-style layer (Geng & Kolter, 2023;

Du & Mordatch, 2019; Liu et al., 2024); TEL replaces this nonlinearity block with a TEL layer with $K$ refinement steps.

**Autoencoder (reconstruction).**    The autoencoder has a symmetric encoder–decoder:

$$x \xrightarrow{W_{\text{enc},1}} h_1 \xrightarrow{\sigma} h_2 \xrightarrow{W_{\text{enc},2}} z \in \mathbb{R}^2, \tag{39}$$

$$z \xrightarrow{W_{\text{dec},1}} \tilde{h}_1 \xrightarrow{\sigma} \tilde{h}_2 \xrightarrow{W_{\text{dec},2}} \hat{x}. \tag{40}$$

TEL replaces the central hidden nonlinearity in both encoder and decoder with a TEL layer, keeping the overall parameter count matched to the MLP baselines.

### G.4.2    STAGE II: LENET, DEEPCONVLSTM, MINILM

**LeNet-5.**    We use a standard LeNet-5 backbone (LeCun et al., 2002) with two convolutional blocks followed by two fully connected (FC) layers.

- **TEL-head:** TEL replaces the *first* FC hidden MLP block *immediately after flattening* (i.e., the first nonlinear projection after the conv part).
- **TEL-full:** TEL replaces both FC hidden MLP blocks, leaving the final classifier layer linear.

**DeepConvLSTM.**    We use the architecture of Ordóñez & Roggen (2016): several 1D convolutions over the temporal dimension followed by stacked LSTM layers and a final classifier MLP.

- **TEL-head:** TEL replaces the *first* feedforward MLP block after the convolutional feature extractor (before the LSTM or classification head, depending on the variant).
- **TEL-full:** TEL replaces all feedforward MLP blocks in the post-convolutional head, keeping the LSTM recurrence and gating mechanisms unchanged.

**MiniLM.**    We use a lightweight Transformer encoder with self-attention and FFN sublayers (MiniLM-style) (Wang et al., 2020).

- **TEL-head:** TEL replaces the *first* FFN sublayer in the encoder stack (i.e., in the first Transformer block).
- **TEL-full:** TEL replaces *every* FFN sublayer in all Transformer blocks.

Attention, positional embeddings, and LayerNorm are unchanged.

### G.4.3    STAGE III: RESNET, VIT, GPT

**ResNet-18 / ResNet-50 (Tiny-ImageNet and ImageNet-1K).**    We follow the standard torchvision implementations of ResNet (He et al., 2016). Each residual block contains a convolutional path and, in bottleneck blocks, an internal "MLP-like" $1 \times 1$ projection. In our experiments we treat the post-activation projection inside the residual unit as the MLP block to be replaced.

- **TEL-head:** TEL replaces the *first* such MLP block in the *first residual unit* of the network (i.e., the first block after the stem). All later residual units remain standard.
- **TEL-full:** TEL replaces the MLP block in *every* residual unit throughout the network, keeping channel dimensions and parameter count matched as closely as possible.

The convolutional stem, downsampling shortcuts, and global average pooling remain unchanged; only the internal MLP-style transformation within the residual units is replaced by TEL.

**ViT-B/32 and ViT-L/32 (COCO-Stuff, Cityscapes).**    We use ViT backbones with patch embedding, multihead self-attention, and FFN blocks (Dosovitskiy, 2020).

- **TEL-head:** TEL replaces the FFN in the *first Transformer block only*.

- **TEL-full:** TEL replaces every FFN block in all Transformer layers.

Patch embeddings, attention blocks, and normalization are unchanged; TEL affects only the FFN sublayers.

**GPT-1 and GPT-2 (LAMBADA, WikiText-2).** We use decoder-only Transformer architectures following GPT-1 and GPT-2 configurations (Radford et al., 2018; 2019).

- **TEL-head:** TEL replaces the FFN in the *first decoder block*.
- **TEL-full:** TEL replaces all FFN sublayers across all decoder blocks.

Attention, positional encodings, and LayerNorm are identical to the baselines; only the FFN nonlinearity is replaced.

### G.5 LOSS FUNCTIONS AND EVALUATION METRICS

TEL does *not* modify the global task loss; the free energy $G_\theta$ is used only to define the layer's internal refinement dynamics. All networks—TEL and baselines—are trained with the same task losses and evaluation metrics.

**Classification losses.** For a classification task with $C$ classes, a model producing logits $f_\theta(x) \in \mathbb{R}^C$, and one-hot label vector $e_y$, we use the standard cross-entropy loss:

$$\mathcal{L}_{\text{CE}}(x, y) = -\sum_{c=1}^{C} \mathbf{1}[y = c] \log p_\theta(c \mid x), \qquad p_\theta(c \mid x) = \frac{\exp(f_\theta(x)_c)}{\sum_{c'=1}^{C} \exp(f_\theta(x)_{c'})}. \tag{41}$$

This is used for all classification tasks: Stage I image classification (MNIST, Fashion-MNIST, CIFAR-10, CIFAR-100, STL-10), Stage II USPS / HAR / AGNews, and Stage III Tiny-ImageNet / ImageNet-1K.

**Regression losses.** For regression tasks with target $y \in \mathbb{R}^d$ and prediction $\hat{y} = f_\theta(x)$, we use mean squared error (MSE):

$$\mathcal{L}_{\text{MSE}}(x, y) = \|f_\theta(x) - y\|_2^2. \tag{42}$$

Evaluation uses root mean squared error (RMSE):

$$\text{RMSE} = \sqrt{\frac{1}{N} \sum_{n=1}^{N} \|f_\theta(x_n) - y_n\|_2^2}. \tag{43}$$

**Reconstruction losses.** For autoencoders with encoder $E_\theta$ and decoder $D_\theta$, the reconstruction loss is

$$\mathcal{L}_{\text{rec}}(x) = \|D_\theta(E_\theta(x)) - x\|_2^2. \tag{44}$$

For reporting in Fig. 2, we use 1/Recon. Error, where Recon. Error is the mean squared reconstruction error on the test set.

**Semantic segmentation losses.** For segmentation with $C$ classes and per-pixel logits $f_\theta(x)_{ij} \in \mathbb{R}^C$, we use per-pixel cross-entropy:

$$\mathcal{L}_{\text{seg}}(x, y) = -\frac{1}{HW} \sum_{i=1}^{H} \sum_{j=1}^{W} \sum_{c=1}^{C} y_{ijc} \log p_\theta(c \mid x)_{ij}, \tag{45}$$

where $H$ and $W$ are height and width, $y_{ijc}$ is the one-hot label, and $p_\theta(c \mid x)_{ij}$ is the softmax of the logits at pixel $(i, j)$. We report mean Intersection-over-Union (mIoU) on the validation and test splits.

**Autoregressive language modeling losses.** For language modeling, given a token sequence $(x_1, \ldots, x_T)$, the model defines conditional probabilities $p_\theta(x_t \mid x_{<t})$. We minimize the negative log-likelihood:

$$\mathcal{L}_{\text{LM}}(x_{1:T}) = -\sum_{t=1}^{T} \log p_\theta(x_t \mid x_{<t}). \tag{46}$$

Perplexity is computed as

$$\text{PPL} = \exp\Big(\frac{1}{T}\sum_{t=1}^{T} -\log p_\theta(x_t \mid x_{<t})\Big). \tag{47}$$

**Evaluation metrics summary.**

- **Accuracy:** used for all classification tasks (Stage I, Stage II, and Stage III image classification).
- **RMSE:** used for all tabular regression benchmarks in Stage I.
- **Reconstruction error / $1/$Recon. Error:** used for Stage I reconstruction tasks.
- **mIoU:** used for semantic segmentation (COCO-Stuff and Cityscapes).
- **Perplexity:** used for language modeling (LAMBADA and WikiText-2).

### G.6 PREPROCESSING AND DATA SPLITS

We follow standard splits and normalization for all datasets, and apply simple data augmentation only for vision tasks where commonly used.

**Splits.**

- **Stage I classification/reconstruction:** official train/test splits (or standard synthetic dataset protocols).
- **Stage I regression:** 20 random splits per dataset, as detailed above.
- **Stage II:** standard splits for USPS, UCI HAR, and AGNews.
- **Stage III:** standard training/validation splits for Tiny-ImageNet, ImageNet-1K, COCO-Stuff, Cityscapes, LAMBADA, and WikiText-2.

**Normalization.**

- **Images:** per-channel mean/std normalization (using dataset statistics).
- **Tabular:** z-score standardization of each feature (zero mean, unit variance).
- **Sequences (HAR):** channel-wise normalization over the training set.
- **Language:** byte-level BPE tokenization (GPT-style) with a fixed vocabulary; tokens are mapped to integer IDs without additional normalization.

**Augmentation.**

- **MNIST / Fashion-MNIST / USPS:** no augmentation.
- **CIFAR-10 / CIFAR-100 / STL-10 / Tiny-ImageNet / ImageNet-1K:** random crop, random horizontal flip; Tiny-ImageNet additionally uses light color jitter.
- **Segmentation (COCO-Stuff, Cityscapes):** scale jitter, random resized crop, horizontal flip.
- **Tabular regression, reconstruction, and language modeling:** no augmentation.

Table 1: Accuracy ($\uparrow$) $\pm$ Std across models. Best per (dataset, hidden size) in **bold**; second-best underlined.

| Dataset | Hidden Size | Linear | MLP | EBM | DEQ | KAN | TEL |
|---|---|---|---|---|---|---|---|
| MNIST | 8 | 0.9231±0.0021 | 0.9244±0.0021 | 0.9261±0.0020 | 0.9316±0.0020 | 0.9274±0.0074 | **0.9367±0.0019** |
| | 16 | 0.9237±0.0015 | 0.9436±0.0015 | 0.9280±0.0016 | 0.9485±0.0016 | 0.9285±0.0050 | **0.9554±0.0015** |
| | 32 | 0.9243±0.0009 | 0.9629±0.0009 | 0.9268±0.0010 | 0.9653±0.0010 | 0.9296±0.0033 | **0.9684±0.0011** |
| | 64 | 0.9262±0.0004 | 0.9635±0.0004 | 0.9267±0.0005 | **0.9801±0.0004** | 0.9249±0.0022 | 0.9789±0.0005 |
| | 128 | 0.9239±0.0006 | 0.9754±0.0006 | 0.9264±0.0007 | 0.9812±0.0007 | 0.9276±0.0026 | **0.9819±0.0006** |
| | 256 | 0.9231±0.0003 | 0.9783±0.0003 | 0.9260±0.0004 | 0.9846±0.0003 | 0.9305±0.0016 | **0.9849±0.0004** |
| | 512 | 0.9218±0.0003 | 0.9813±0.0003 | 0.9267±0.0003 | 0.9859±0.0003 | 0.9268±0.0014 | **0.9883±0.0003** |
| | 1024 | 0.9219±0.0003 | 0.9812±0.0002 | 0.9259±0.0004 | **0.9860±0.0002** | 0.9270±0.0011 | **0.9860±0.0002** |
| FashionMNIST | 8 | 0.8416±0.0015 | 0.8432±0.0015 | 0.8454±0.0016 | 0.8443±0.0016 | 0.8535±0.0048 | **0.8532±0.0015** |
| | 16 | 0.8397±0.0014 | 0.8540±0.0014 | 0.8460±0.0015 | 0.8577±0.0014 | 0.8567±0.0045 | **0.8668±0.0013** |
| | 32 | 0.8379±0.0013 | 0.8649±0.0013 | 0.8461±0.0013 | 0.8712±0.0014 | 0.8598±0.0050 | **0.8762±0.0015** |
| | 64 | 0.8416±0.0024 | 0.8818±0.0024 | 0.8458±0.0023 | 0.8877±0.0025 | 0.8608±0.0063 | **0.8902±0.0024** |
| | 128 | 0.8405±0.0014 | 0.8846±0.0014 | 0.8445±0.0015 | 0.8917±0.0013 | 0.8603±0.0061 | **0.8942±0.0014** |
| | 256 | 0.8425±0.0025 | 0.8842±0.0025 | 0.8432±0.0022 | 0.8934±0.0025 | 0.8508±0.0074 | **0.8991±0.0026** |
| | 512 | 0.8390±0.0012 | 0.8897±0.0012 | 0.8415±0.0013 | 0.9012±0.0012 | 0.8525±0.0035 | **0.9102±0.0011** |
| | 1024 | 0.8387±0.0010 | 0.8895±0.0012 | 0.8417±0.0011 | 0.9082±0.0011 | 0.8526±0.0043 | **0.9108±0.0010** |
| CIFAR10 | 8 | 0.3991±0.0019 | 0.4012±0.0019 | 0.4051±0.0020 | **0.4147±0.0018** | 0.4119±0.0051 | 0.4137±0.0019 |
| | 16 | 0.3995±0.0023 | 0.4345±0.0023 | 0.4100±0.0024 | 0.4434±0.0024 | 0.4073±0.0061 | **0.4597±0.0025** |
| | 32 | 0.3999±0.0026 | 0.4679±0.0026 | 0.4114±0.0025 | 0.4722±0.0026 | 0.4028±0.0083 | **0.4964±0.0024** |
| | 64 | 0.4028±0.0021 | 0.4890±0.0021 | 0.4104±0.0020 | 0.4935±0.0022 | 0.4103±0.0079 | **0.5128±0.0021** |
| | 128 | 0.4006±0.0024 | 0.5128±0.0024 | 0.4091±0.0023 | 0.5144±0.0023 | 0.4025±0.0075 | **0.5227±0.0022** |
| | 256 | 0.4025±0.0018 | 0.5147±0.0018 | 0.4078±0.0017 | 0.5228±0.0018 | 0.3965±0.0045 | **0.5325±0.0019** |
| | 512 | 0.3870±0.0026 | 0.5203±0.0026 | 0.4040±0.0027 | 0.5327±0.0028 | 0.3957±0.0092 | **0.5371±0.0027** |
| | 1024 | 0.3872±0.0027 | 0.5205±0.0027 | 0.4003±0.0026 | 0.5326±0.0028 | 0.3956±0.0079 | **0.5451±0.0029** |
| CIFAR100 | 8 | 0.1368±0.0014 | 0.1271±0.0014 | 0.1400±0.0015 | 0.1394±0.0015 | 0.1266±0.0047 | **0.1441±0.0016** |
| | 16 | 0.1538±0.0015 | 0.1588±0.0015 | 0.1670±0.0016 | 0.1728±0.0017 | 0.1528±0.0045 | **0.1783±0.0017** |
| | 32 | 0.1708±0.0017 | 0.1906±0.0017 | 0.1734±0.0018 | 0.2063±0.0018 | 0.1790±0.0053 | **0.2075±0.0019** |
| | 64 | 0.1723±0.0020 | 0.2102±0.0020 | 0.1765±0.0021 | 0.2227±0.0020 | 0.1876±0.0077 | **0.2284±0.0021** |
| | 128 | 0.1613±0.0016 | 0.2223±0.0016 | 0.1746±0.0017 | 0.2307±0.0015 | 0.1827±0.0039 | **0.2390±0.0014** |
| | 256 | 0.1640±0.0011 | 0.2307±0.0011 | 0.1727±0.0012 | 0.2380±0.0012 | 0.1807±0.0040 | **0.2495±0.0013** |
| | 512 | 0.1620±0.0019 | 0.2344±0.0019 | 0.1657±0.0020 | 0.2467±0.0021 | 0.1769±0.0053 | **0.2532±0.0022** |
| | 1024 | 0.1622±0.0019 | 0.2342±0.0017 | 0.1626±0.0018 | 0.2470±0.0019 | 0.1771±0.0053 | **0.2591±0.0018** |
| STL10 | 8 | 0.3396±0.0047 | 0.3108±0.0047 | 0.3403±0.0048 | 0.3568±0.0050 | **0.3785±0.0152** | 0.3343±0.0051 |
| | 16 | 0.3395±0.0038 | 0.3439±0.0038 | 0.3634±0.0037 | 0.3716±0.0038 | **0.3810±0.0116** | 0.3624±0.0039 |
| | 32 | 0.3394±0.0029 | 0.3771±0.0029 | 0.3498±0.0030 | **0.3863±0.0031** | 0.3836±0.0115 | 0.3806±0.0032 |
| | 64 | 0.3288±0.0022 | 0.3899±0.0022 | 0.3499±0.0023 | 0.3922±0.0022 | 0.3719±0.0081 | **0.3991±0.0023** |
| | 128 | 0.3149±0.0025 | 0.3939±0.0025 | 0.3417±0.0026 | 0.4014±0.0026 | 0.3779±0.0091 | **0.4096±0.0027** |
| | 256 | 0.2988±0.0014 | 0.4060±0.0014 | 0.3335±0.0015 | 0.4068±0.0013 | 0.3784±0.0061 | **0.4141±0.0014** |
| | 512 | 0.3005±0.0027 | 0.4024±0.0027 | 0.3333±0.0026 | 0.4146±0.0027 | 0.3681±0.0086 | **0.4178±0.0028** |
| | 1024 | 0.3007±0.0028 | 0.4023±0.0028 | 0.3334±0.0029 | 0.4146±0.0030 | 0.3683±0.0075 | **0.4183±0.0029** |

## G.7 HARDWARE

Stages I–II (shallow and mid-scale models) are trained and evaluated on a single workstation with an RTX 6000 Ada GPU, a 16-core CPU, and 64 GB RAM. Stage III (large-scale ResNet/ViT/GPT models) is trained on a cluster with 4×NVIDIA A100 GPUs, 64 CPU cores, and 128 GB RAM, but all reported runtime and latency measurements are taken on the RTX 6000 Ada workstation during inference.

For latency measurements, we use batch size 1, discard 500 warmup iterations, and average over the next 1000 iterations. TEL, MLP, DEQ, EBM, and KAN baselines use identical batch sizes, FLOP budgets, memory budgets, and hidden dimensions wherever they are compared. FLOPs are computed with a combination of analytical formulas for linear and convolutional layers and counting TEL's refinement as $K$ additional elementwise passes over the hidden representation, as detailed in Appendix H.3.

## H ADDITIONAL RESULTS AND INSIGHTS

### H.1 DIMENSIONALITY COMPARISON

Tables 1, 2, and 3 summarize average classification accuracy, regression RMSE, and reconstruction error (mean ± std) over repeated runs and seeds across the above datasets. We sweep hidden dimensions from 8 to 1024. At matched hidden size, TEL leads on the majority of tasks while exhibiting equal or lower variability across seeds. Across 15 datasets, TEL delivers higher accuracy or lower error than strong baselines at matched capacity (5/5 classification; 4/5 regression; 5/5 reconstruction) and comparable or lower run-to-run variability, with especially large stability gains on reconstruc-

Table 2: RMSE ($\downarrow$) $\pm$ Std across models. Best (lowest) per (dataset, hidden size) in **bold**; second-best underlined.

| Dataset | Hidden Size | Linear | MLP | EBM | DEQ | KAN | TEL |
|---|---|---|---|---|---|---|---|
| Diabetes | 8 | 57.4949±0.3830 | 58.4617±0.4036 | 59.3100±0.3950 | 57.9041±0.3600 | 61.4805±0.8298 | **56.1183±0.3703** |
| | 16 | 57.4693±0.4911 | 57.4000±0.3728 | 59.2949±0.4700 | 56.7867±0.3300 | 60.7735±0.7719 | **56.0947±0.3185** |
| | 32 | 57.4503±0.4681 | 56.7506±0.3525 | 59.2836±0.4520 | 57.2521±0.3100 | 60.0151±0.5628 | **56.0200±0.3137** |
| | 64 | 57.4281±0.4033 | 56.6082±0.4853 | 59.2695±0.4200 | 58.2355±0.3450 | 59.4916±0.7164 | **55.9541±0.3533** |
| | 128 | 57.4177±0.4898 | 56.6344±0.3388 | 59.2527±0.4780 | 58.6273±0.3800 | 59.1089±0.7658 | **55.9195±0.3873** |
| | 256 | 57.4073±0.4957 | 56.6607±0.3887 | 59.2359±0.4890 | 59.0192±0.4000 | 58.7263±0.8066 | **55.8848±0.4045** |
| | 512 | 57.3915±0.4137 | 56.7422±0.4511 | 59.2150±0.4250 | 59.4898±0.4200 | 58.5837±0.8055 | **55.9421±0.4149** |
| | 1024 | 57.3694±0.4279 | 56.9094±0.4942 | 59.1987±0.4400 | 59.7696±0.3500 | 58.4132±0.9567 | **56.0335±0.3625** |
| Energy | 8 | 2.8261±0.2235 | 2.6963±0.2115 | 2.7734±0.2150 | 2.5827±0.1650 | **0.9885±0.1288** | 1.6912±0.1706 |
| | 16 | 2.8354±0.2213 | 2.2565±0.1477 | 2.7786±0.2200 | 1.4903±0.0400 | 0.9891±0.0702 | **0.8441±0.0364** |
| | 32 | 2.8377±0.1950 | 1.1616±0.0413 | 2.7744±0.1900 | 0.5270±0.0270 | 0.9891±0.0458 | **0.4448±0.0258** |
| | 64 | 2.8291±0.2188 | 0.5602±0.0441 | 2.7756±0.2100 | 0.4597±0.0320 | 0.9875±0.0613 | **0.4217±0.0306** |
| | 128 | 2.8200±0.1943 | 0.4746±0.0292 | 2.7658±0.2000 | 0.4558±0.0280 | 0.9882±0.0591 | **0.3793±0.0270** |
| | 256 | 2.8108±0.2641 | 0.3890±0.0271 | 2.7559±0.2550 | 0.4519±0.0250 | 0.9889±0.0522 | **0.3370±0.0236** |
| | 512 | 2.8187±0.2299 | 0.3687±0.0232 | 2.7597±0.2350 | 0.4775±0.0240 | 0.9849±0.0504 | **0.3370±0.0247** |
| | 1024 | 2.7942±0.2523 | 0.3653±0.0283 | 2.7399±0.2400 | 0.5074±0.0260 | 0.9786±0.0605 | **0.3307±0.0264** |
| Concrete | 8 | 10.5903±0.8138 | 8.7529±0.7301 | 10.0277±0.8000 | 7.6779±0.6200 | 6.2085±0.7789 | **5.6212±0.6167** |
| | 16 | 10.5817±0.7901 | 8.1200±0.6533 | 10.0268±0.7700 | 6.6732±0.5200 | 6.1939±0.7608 | **5.5128±0.5113** |
| | 32 | 10.5849±0.7934 | 6.9765±0.5187 | 10.0238±0.7800 | 5.6741±0.3100 | 6.1946±0.5696 | **5.1550±0.3086** |
| | 64 | 10.5744±1.0725 | 5.9690±0.4754 | 10.0239±1.0200 | 5.3430±0.3150 | 6.1985±0.8572 | **4.6646±0.3084** |
| | 128 | 10.5737±0.7713 | 5.5254±0.4681 | 10.0206±0.7600 | 5.0815±0.3000 | 6.1978±0.6927 | **4.4251±0.3023** |
| | 256 | 10.5730±0.7509 | 5.0817±0.3899 | 10.0173±0.7400 | 4.8200±0.3100 | 6.1972±0.6680 | **4.1855±0.3051** |
| | 512 | 10.5693±0.9758 | 4.7136±0.3203 | 10.0138±0.9500 | 4.4154±0.2700 | 6.1814±0.4876 | **3.8300±0.2612** |
| | 1024 | 10.5695±0.9177 | 4.5960±0.3511 | 10.0192±0.9000 | 4.2742±0.3800 | 6.1806±0.6942 | **3.5965±0.3899** |
| Wine | 8 | 0.5984±0.0045 | 0.6095±0.0039 | 0.6392±0.0045 | 0.6849±0.0042 | 0.6288±0.0081 | **0.5831±0.0044** |
| | 16 | 0.5978±0.0043 | 0.6048±0.0051 | 0.6390±0.0044 | 0.6481±0.0040 | 0.6213±0.0086 | **0.5858±0.0041** |
| | 32 | 0.5977±0.0050 | 0.6050±0.0036 | 0.6389±0.0050 | 0.6522±0.0041 | 0.6215±0.0077 | **0.5816±0.0042** |
| | 64 | 0.5962±0.0060 | 0.5977±0.0050 | 0.6385±0.0060 | 0.6496±0.0047 | 0.6210±0.0091 | **0.5791±0.0048** |
| | 128 | 0.5956±0.0042 | 0.5895±0.0043 | 0.6380±0.0042 | 0.6428±0.0030 | 0.6198±0.0056 | **0.5719±0.0030** |
| | 256 | 0.5951±0.0050 | 0.5812±0.0054 | 0.6375±0.0049 | 0.6360±0.0042 | 0.6186±0.0073 | **0.5647±0.0042** |
| | 512 | 0.5953±0.0048 | 0.5671±0.0047 | 0.6357±0.0047 | 0.6464±0.0038 | 0.6221±0.0057 | **0.5507±0.0038** |
| | 1024 | 0.5937±0.0044 | 0.5549±0.0041 | 0.6359±0.0045 | 0.6875±0.0043 | 0.6197±0.0108 | **0.5388±0.0042** |
| California | 8 | 0.7348±0.0110 | 0.5792±0.0084 | 0.6963±0.0110 | **0.5448±0.0071** | 0.6238±0.0141 | 0.5655±0.0071 |
| | 16 | 0.7351±0.0121 | 0.5549±0.0089 | 0.6962±0.0120 | **0.4916±0.0076** | 0.6266±0.0159 | 0.5488±0.0076 |
| | 32 | 0.7349±0.0107 | 0.5432±0.0076 | 0.6961±0.0105 | **0.4674±0.0067** | 0.6239±0.0129 | 0.5416±0.0066 |
| | 64 | 0.7351±0.0129 | 0.5425±0.0093 | 0.6963±0.0128 | **0.4888±0.0077** | 0.6240±0.0164 | 0.5373±0.0076 |
| | 128 | 0.7350±0.0112 | 0.5371±0.0079 | 0.6965±0.0112 | **0.4908±0.0067** | 0.6238±0.0127 | 0.5294±0.0067 |
| | 256 | 0.7349±0.0118 | 0.5317±0.0083 | 0.6967±0.0117 | **0.4929±0.0070** | 0.6236±0.0143 | 0.5215±0.0070 |
| | 512 | 0.7362±0.0113 | 0.5209±0.0078 | 0.6967±0.0114 | **0.4745±0.0066** | 0.6238±0.0133 | 0.5176±0.0066 |
| | 1024 | 0.7353±0.0127 | 0.5257±0.0089 | 0.6967±0.0126 | **0.4829±0.0080** | 0.6256±0.0177 | 0.5193±0.0081 |

tion. These empirical trends align with TEL's iterative refinement and adaptive temperature, which together provide reliable optimization dynamics even at higher widths.

## H.2 PERFORMANCE AT MATCHED PARAMS, AND FLOPs

Figure 8 compares all six architectures in the joint space of parameter count, FLOPs, and downstream performance. Each point corresponds to a specific hidden size (8–1028), averaged over 20 runs and 5 random seeds across the 15 datasets. Across this Pareto frontier, TEL consistently occupies the upper-left region, attaining higher accuracy or lower error at comparable or lower computational cost than strong nonlinear baselines such as MLP, EBM, DEQ, and KAN. In particular, TEL typically matches the best-performing competitors in terms of raw metric while requiring fewer parameters or FLOPs, indicating a more favorable performance–efficiency trade-off. This pattern holds robustly across seeds, suggesting that TEL's iterative refinement and adaptive temperature not only improve absolute performance but also deliver these gains without incurring additional computational overhead.

## H.3 RUNTIME AND MEMORY SCALING WITH $K$

We evaluate the empirical runtime behavior of TEL as a function of the refinement depth $K$, complementing the FLOP characterization in §2.3. Although the theoretical cost scales linearly in $K$ due to the unrolled refinement equation 8, modern GPU runtimes can deviate from FLOP counts because of kernel fusion, launch overhead, and memory traffic. Accordingly, we report wall-clock throughput and memory from end-to-end forward+backward passes in Table 4.

Table 3: Reconstruction error ($\downarrow$) $\pm$ Std across models. Best (lowest) per (dataset, hidden size) in **bold**; second-best underlined.

| Dataset | Hidden Size | Linear | MLP | EBM | DEQ | KAN | TEL |
|---|---|---|---|---|---|---|---|
| Sinusoid 1D | 8 | 0.2583±0.029 | 0.2617±0.014 | 0.2207±0.028 | 0.2680±0.0145 | 0.3775±0.047 | **0.2096±0.015** |
| | 16 | 0.2523±0.028 | 0.2548±0.013 | 0.2529±0.027 | 0.2605±0.0138 | 0.3647±0.045 | **0.2374±0.014** |
| | 32 | 0.2553±0.027 | 0.2489±0.012 | 0.2485±0.026 | 0.2778±0.0136 | 0.3685±0.046 | **0.2277±0.014** |
| | 64 | 0.2600±0.028 | 0.2703±0.014 | 0.2230±0.027 | 0.2495±0.0132 | 0.3500±0.043 | **0.2118±0.013** |
| | 128 | 0.2322±0.025 | 0.2861±0.015 | 0.2493±0.024 | 0.2734±0.0122 | 0.3342±0.041 | **0.2048±0.012** |
| | 256 | 0.2475±0.027 | 0.2998±0.016 | 0.2246±0.026 | 0.2495±0.0131 | 0.3681±0.047 | **0.2134±0.013** |
| | 512 | 0.2751±0.031 | 0.3125±0.017 | 0.2506±0.030 | 0.4518±0.0158 | 0.3641±0.045 | **0.2381±0.016** |
| | 1024 | 0.3204±0.036 | 0.3416±0.019 | 0.2276±0.035 | 0.6240±0.0185 | 0.4336±0.057 | **0.2162±0.019** |
| Moons 2D | 8 | 0.3406±0.019 | 0.3541±0.009 | 0.3262±0.0185 | 0.3475±0.0098 | 0.3540±0.023 | **0.3099±0.010** |
| | 16 | 0.3409±0.019 | 0.3616±0.009 | 0.3339±0.0188 | 0.3312±0.0093 | 0.3483±0.022 | **0.3143±0.0095** |
| | 32 | 0.3359±0.0185 | 0.3627±0.009 | 0.3205±0.0182 | 0.3323±0.0094 | 0.3628±0.023 | **0.3044±0.0095** |
| | 64 | 0.3455±0.019 | 0.3577±0.009 | 0.3119±0.0189 | 0.3216±0.0089 | 0.3622±0.023 | **0.2963±0.009** |
| | 128 | 0.3677±0.0205 | 0.3473±0.0085 | 0.3112±0.0202 | 0.3371±0.0097 | 0.3467±0.022 | **0.2956±0.010** |
| | 256 | 0.3856±0.0215 | 0.3446±0.0085 | 0.3206±0.021 | 0.3473±0.0103 | 0.3480±0.0225 | **0.3062±0.0105** |
| | 512 | 0.3995±0.0225 | 0.3517±0.009 | 0.3319±0.022 | 0.3847±0.0102 | 0.3681±0.024 | **0.3153±0.0105** |
| | 1024 | 0.4076±0.0235 | 0.3623±0.0095 | 0.3455±0.023 | 0.4244±0.0108 | 0.3771±0.0245 | **0.3212±0.011** |
| Spiral 2D | 8 | 0.6279±0.073 | 0.5513±0.028 | 0.5251±0.071 | 0.5099±0.031 | 10.4261±1.39 | **0.4999±0.032** |
| | 16 | 0.6486±0.075 | 0.5088±0.026 | 0.4958±0.073 | 0.5335±0.0285 | 10.5825±1.44 | **0.4710±0.029** |
| | 32 | 0.5588±0.064 | 0.5687±0.030 | 0.4445±0.062 | 0.5371±0.0275 | 10.3620±1.35 | **0.4223±0.028** |
| | 64 | 0.6466±0.075 | 0.5363±0.027 | 0.4462±0.073 | 0.5042±0.0295 | 11.0432±1.45 | **0.4239±0.030** |
| | 128 | 0.6806±0.078 | 0.5714±0.031 | 0.5730±0.076 | 0.5561±0.032 | 10.5546±1.37 | **0.5413±0.033** |
| | 256 | 0.9773±0.112 | 0.5238±0.026 | 0.6066±0.108 | 0.6176±0.029 | 11.8035±1.59 | **0.4971±0.030** |
| | 512 | 1.1061±0.129 | 0.4831±0.024 | 0.5587±0.124 | 0.8603±0.027 | 10.8812±1.42 | **0.4559±0.028** |
| | 1024 | 2.6649±0.323 | **0.6838±0.037** | 0.7369±0.310 | 1.1374±0.038 | 12.0361±1.76 | 0.6971±0.039 |
| Swissroll 3D | 8 | 13.3184±1.49 | 13.3449±0.72 | 3.7975±0.145 | 13.0688±0.62 | 15.2981±1.99 | **3.6076±0.063** |
| | 16 | 13.3565±1.50 | 13.3506±0.73 | 2.8974±0.148 | 13.0265±0.67 | 15.2440±1.95 | **2.7333±0.068** |
| | 32 | 13.3426±1.48 | 13.4280±0.74 | 2.5612±0.146 | 13.0468±0.78 | 15.3001±1.99 | **2.4331±0.080** |
| | 64 | 13.3051±1.47 | 13.4027±0.73 | 2.0404±0.144 | 13.1222±0.64 | 15.3607±2.00 | **1.9383±0.065** |
| | 128 | 13.3503±1.51 | 13.4676±0.75 | 1.7073±0.149 | 13.2145±0.68 | 15.9867±2.08 | **1.3890±0.069** |
| | 256 | 13.7316±1.57 | 13.3840±0.73 | 2.0952±0.155 | 13.1223±0.70 | 16.0167±2.12 | **1.8464±0.071** |
| | 512 | 13.8267±1.59 | 13.4397±0.74 | 2.1415±0.157 | 13.2337±0.60 | 15.8926±2.07 | **1.8105±0.061** |
| | 1024 | 15.9496±1.89 | 13.4512±0.76 | 1.9840±0.185 | 13.7061±0.83 | 18.1510±2.37 | **1.8848±0.085** |
| Spheres 3D | 8 | 0.3230±0.035 | 0.3167±0.016 | 0.1684±0.034 | 0.3180±0.0125 | 0.3178±0.031 | **0.1600±0.012** |
| | 16 | 0.2965±0.032 | 0.3180±0.016 | 0.1912±0.031 | 0.3191±0.0132 | 0.3241±0.042 | **0.1816±0.013** |
| | 32 | 0.3188±0.035 | 0.3123±0.015 | 0.2021±0.034 | 0.3151±0.0115 | 0.3310±0.033 | **0.1790±0.011** |
| | 64 | 0.3363±0.037 | 0.3213±0.016 | **0.1535±0.036** | 0.3394±0.0108 | 0.3541±0.046 | 0.1858±0.011 |
| | 128 | 0.3172±0.035 | 0.2982±0.015 | 0.2222±0.034 | 0.3105±0.0112 | 0.3492±0.035 | **0.1901±0.011** |
| | 256 | 0.3278±0.038 | 0.3348±0.017 | 0.2437±0.037 | 0.3350±0.0128 | 0.3202±0.031 | **0.2015±0.013** |
| | 512 | 0.3171±0.035 | 0.3054±0.015 | 0.2206±0.034 | 0.4104±0.0112 | 0.3492±0.045 | **0.2095±0.011** |
| | 1024 | 0.3658±0.043 | 0.3591±0.019 | **0.1660±0.042** | 0.5838±0.0155 | 0.3482±0.025 | 0.2177±0.016 |

All measurements use batch size 512 and hidden dimension 256. We compare: (i) an MLP baseline with ReLU activations; (ii) TEL with $K \in \{1, 3, 5, 7, 10\}$; (iii) an EBM-style energy refinement; and (iv) a DEQ model with Anderson acceleration. All numbers are averaged over 20 warm-started runs on a single RTX 6000 Ada GPU.

The empirical scaling matches the theoretical structure in §2.3: TEL's runtime grows linearly with $K$ because each refinement step applies the same non-expansive update map equation 8. In the practical range $K \in \{3, 5\}$ that yields the best accuracy–latency tradeoff (§3), TEL is only $1.3\times$–$1.9\times$ slower than an MLP of identical width, while being significantly faster than DEQ and less memory-intensive than KAN.

Peak memory increases modestly due to storing the $K$ intermediate states; however, enabling checkpointing (§2.2) reduces this to nearly constant memory in practice.

## H.4 Temperature Update

We analyze how different temperature–update strategies impact TEL's performance. Figure 9 compares a fixed temperature $T$ with several adaptive schedules $T_t$ at hidden dimension 256. We consider two estimators for the adaptive temperature: (i) a Gaussian-based estimator and (ii) a two-layer MLP estimator, each instantiated in both global and channel-wise variants. Across all configurations, adaptive temperatures $T_t$ consistently outperform a fixed $T$, indicating that allowing the refinement dynamics to adjust their sharpness over iterations reliably improves optimization and accuracy. The MLP-based estimator yields the most stable behavior, remaining robust across batch sizes, while the global Gaussian estimator attains the highest peak accuracy with only a negligible increase in

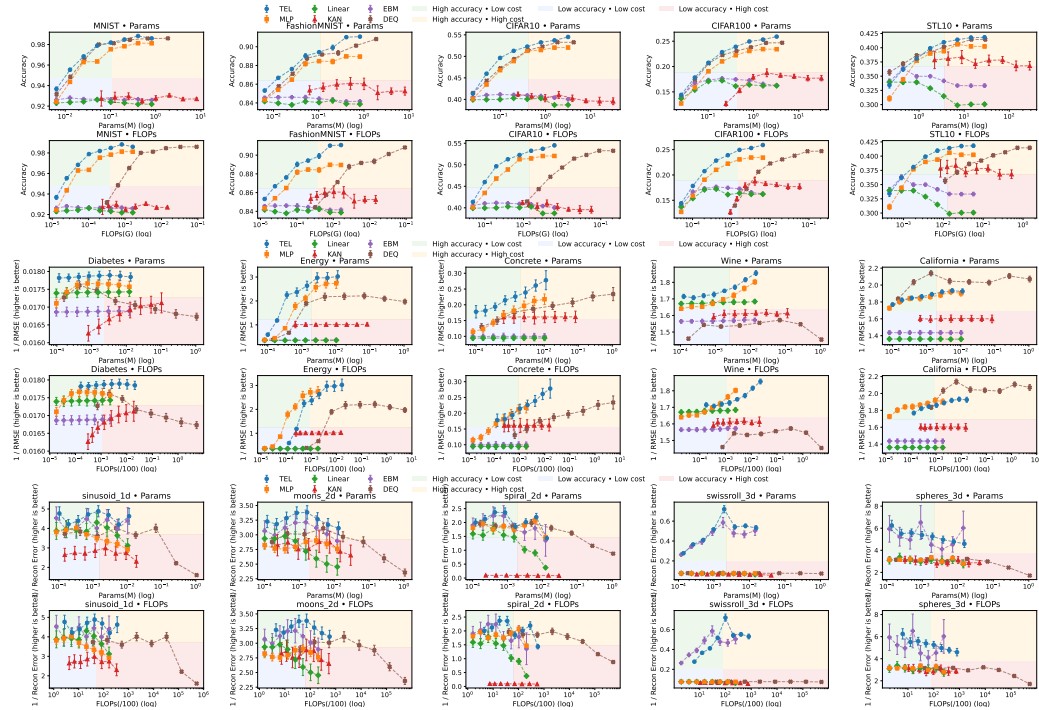

Figure 8: Average performance ($\pm$ std) over 20 runs across 5 different random seeds, evaluated on 15 datasets using six building-block models: Linear, MLP (Linear+ReLU), KAN, EBM, DEQ, and TEL for hidden embedding size ranging from 8 to 1028 plotted against parameter count and FLOPs.

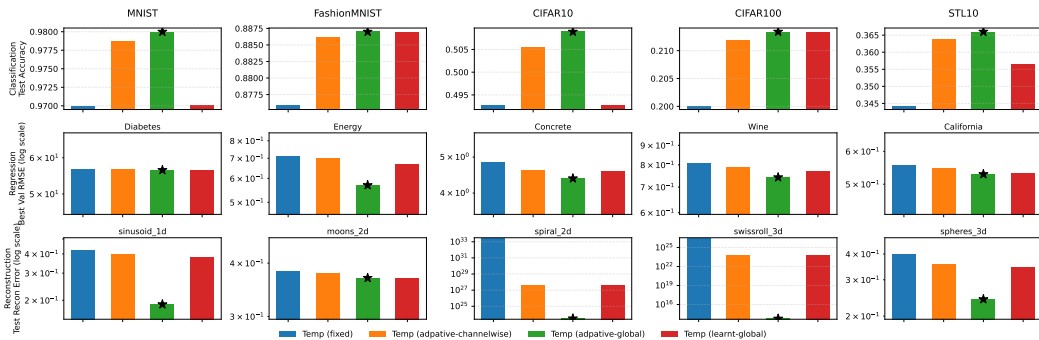

Figure 9: Temperature–update ablation at hidden dimension 256. We compare a fixed temperature $T$ against adaptive schedules $T_t$ using either a Gaussian-based estimator or a two-layer MLP estimator, each in global and channel-wise variants. Adaptive $T_t$ consistently improves performance over fixed $T$; the MLP estimator is the most stable across batch sizes, while the global estimator achieves the highest peak accuracy with minimal additional parameter cost.

parameter count. Overall, TEL is robust to the specific estimator choice, but benefits substantially from using an adaptive rather than fixed temperature.

## H.5 ADDITIONAL COMPARISON: STAGE II

Table 5 reports inference-time costs for the medium-scale TEL benchmarks used in Stage II. For each backbone model, we compare three variants: the original (Vanilla) architecture, a TEL head

Table 4: Runtime and memory scaling with refinement depth $K$. All timing values are reported per sample (converted from per-batch measurements). TEL scales linearly with $K$ and remains substantially cheaper than DEQ or KAN. TEL with $K=1$ is consistently the second-fastest method after the MLP baseline.

| Dataset / Method | $K$ | Time (ms/sample) | Throughput (samples/s) | Peak Mem (GB) |
|---|---|---|---|---|
| CIFAR10 / Classification | | | | |
| MLP (ReLU) | – | 0.00016 | $1.17 \times 10^7$ | 0.018 |
| TEL | 1 | 0.00051 | $3.84 \times 10^6$ | 0.020 |
| TEL | 3 | 0.00113 | $1.75 \times 10^6$ | 0.020 |
| TEL | 5 | 0.00172 | $1.16 \times 10^6$ | 0.020 |
| TEL | 10 | 0.00328 | $6.13 \times 10^5$ | 0.020 |
| EBM | – | 0.00133 | $1.51 \times 10^6$ | 0.019 |
| DEQ | – | 0.00285 | $6.99 \times 10^5$ | 0.025 |
| KAN | – | 0.00176 | $1.13 \times 10^6$ | 0.056 |
| California Housing / Regression | | | | |
| MLP (ReLU) | – | 0.00012 | $1.82 \times 10^7$ | 0.009 |
| TEL | 1 | 0.00063 | $3.27 \times 10^6$ | 0.011 |
| TEL | 3 | 0.00195 | $1.02 \times 10^6$ | 0.011 |
| TEL | 5 | 0.00320 | $6.21 \times 10^5$ | 0.011 |
| TEL | 10 | 0.00609 | $3.28 \times 10^5$ | 0.011 |
| EBM | – | 0.00137 | $1.48 \times 10^6$ | 0.011 |
| DEQ | – | 0.00633 | $3.17 \times 10^5$ | 0.011 |
| KAN | – | 0.00113 | $1.75 \times 10^6$ | 0.032 |
| SwissRoll / Reconstruction | | | | |
| MLP (ReLU) | – | 0.00012 | $1.92 \times 10^7$ | 0.009 |
| TEL | 1 | 0.00055 | $3.76 \times 10^6$ | 0.011 |
| TEL | 3 | 0.00117 | $1.73 \times 10^6$ | 0.011 |
| TEL | 5 | 0.00176 | $1.14 \times 10^6$ | 0.011 |
| TEL | 10 | 0.00324 | $6.20 \times 10^5$ | 0.011 |
| EBM | – | 0.00129 | $1.54 \times 10^6$ | 0.011 |
| DEQ | – | 0.00410 | $4.87 \times 10^5$ | 0.011 |
| KAN | – | 0.00109 | $1.85 \times 10^6$ | 0.035 |

Table 5: Inference cost for the medium-size TEL benchmarks.

| Model | Variant | Params | FLOPs | Latency |
|---|---|---|---|---|
| USPS | | | | |
| | Vanilla | $8.28 \times 10^4$ | $1.24 \times 10^6$ | 0.10 |
| LeNet | TEL head | $8.30 \times 10^4$ | $1.25 \times 10^6$ | 0.12 |
| | TEL full | $8.31 \times 10^4$ | $1.25 \times 10^6$ | 2.45 |
| UCI-HAR | | | | |
| | Vanilla | $2.72 \times 10^5$ | $6.60 \times 10^7$ | 0.55 |
| DeepConvLSTM | TEL head | $2.72 \times 10^5$ | $6.60 \times 10^7$ | 0.66 |
| | TEL full | $2.72 \times 10^5$ | $6.60 \times 10^7$ | 1.04 |
| | Vanilla | $3.34 \times 10^7$ | $5.45 \times 10^9$ | 1.96 |
| MiniLM | TEL head | $3.34 \times 10^7$ | $5.45 \times 10^9$ | 2.07 |
| | TEL full | $3.35 \times 10^7$ | $5.45 \times 10^9$ | 2.81 |

applied on top of frozen features, and a full TEL-equipped model where every block is replaced by its TEL counterpart. Across USPS, UCI-HAR, and AGNews, the TEL head introduces only a small increase in latency while keeping the parameter and FLOP budgets essentially unchanged. The full TEL variant is slower, as expected from its iterative refinement, but remains within practical inference cost ranges for all three tasks.

## H.6 FULL COMPARISON STAGE III

Table 6 summarizes inference costs for the large-scale TEL benchmarks used in Stage III. We evaluate TEL in two configurations applied only at the output head (TEL head) or integrated throughout the entire architecture (TEL full) and compare both variants against the original backbone model. Results are reported across diverse modalities and model families, including convolutional networks (ResNet-18/50), vision transformers (ViT-B/L), and GPT-2 language models, spanning classification, segmentation, and auto-regressive text generation.

Table 6: Inference cost for the large TEL benchmarks.

| Dataset | Task | Model | Variant | Params | FLOPs | Latency |
|---------|------|-------|---------|--------|-------|---------|
| Tiny-ImageNet | Classification | ResNet-18 | Vanilla | $1.13 \times 10^7$ | $2.98 \times 10^8$ | 0.69 |
| | | | TEL head | $1.15 \times 10^7$ | $2.99 \times 10^8$ | 0.73 |
| | | | TEL full | $1.15 \times 10^7$ | $2.99 \times 10^8$ | 1.54 |
| | | ResNet-50 | Vanilla | $2.39 \times 10^7$ | $6.75 \times 10^8$ | 1.74 |
| | | | TEL head | $2.81 \times 10^7$ | $6.84 \times 10^8$ | 1.89 |
| | | | TEL full | $2.81 \times 10^7$ | $6.84 \times 10^8$ | 2.58 |
| ImageNet-1K | Classification | ResNet-18 | Vanilla | $1.17 \times 10^7$ | $3.65 \times 10^9$ | 0.70 |
| | | | TEL head | $1.20 \times 10^7$ | $3.65 \times 10^9$ | 0.74 |
| | | | TEL full | $1.20 \times 10^7$ | $3.65 \times 10^9$ | 1.54 |
| | | ResNet-50 | Vanilla | $2.56 \times 10^7$ | $8.26 \times 10^9$ | 1.72 |
| | | | TEL head | $2.98 \times 10^7$ | $8.27 \times 10^9$ | 1.89 |
| | | | TEL full | $2.98 \times 10^7$ | $8.27 \times 10^9$ | 2.59 |
| COCO-Stuff | Segmentation | ViT-B/32 | Vanilla | $9.38 \times 10^7$ | $5.03 \times 10^{10}$ | 3.28 |
| | | | TEL head | $9.44 \times 10^7$ | $5.06 \times 10^{10}$ | 3.45 |
| | | | TEL full | $9.91 \times 10^7$ | $5.79 \times 10^{10}$ | 3.89 |
| | | ViT-L/32 | Vanilla | $3.16 \times 10^8$ | $1.69 \times 10^{11}$ | 9.79 |
| | | | TEL head | $3.17 \times 10^8$ | $1.69 \times 10^{11}$ | 9.84 |
| | | | TEL full | $3.68 \times 10^8$ | $1.94 \times 10^{11}$ | 10.26 |
| Cityscapes | Segmentation | ViT-B/32 | Vanilla | $9.37 \times 10^7$ | $5.02 \times 10^{10}$ | 3.30 |
| | | | TEL head | $9.43 \times 10^7$ | $5.05 \times 10^{10}$ | 3.55 |
| | | | TEL full | $9.90 \times 10^7$ | $5.77 \times 10^{10}$ | 3.96 |
| | | ViT-L/32 | Vanilla | $3.16 \times 10^8$ | $1.68 \times 10^{11}$ | 10.15 |
| | | | TEL head | $3.17 \times 10^8$ | $1.69 \times 10^{11}$ | 10.18 |
| | | | TEL full | $3.68 \times 10^8$ | $1.94 \times 10^{11}$ | 10.29 |
| WikiText-2 | Auto-regression | GPT-1, GPT-2 | Vanilla | $1.24 \times 10^8$ | $1.27 \times 10^{11}$ | 5.85 |
| | | | TEL head | $1.25 \times 10^8$ | $1.27 \times 10^{11}$ | 7.09 |
| | | | TEL full | $1.64 \times 10^8$ | $1.27 \times 10^{11}$ | 17.15 |
| LAMBADA | Auto-regression | GPT-1, GPT-2 | Vanilla | $1.24 \times 10^8$ | $1.27 \times 10^{11}$ | 5.85 |
| | | | TEL head | $1.25 \times 10^8$ | $1.27 \times 10^{11}$ | 7.09 |
| | | | TEL full | $1.64 \times 10^8$ | $1.27 \times 10^{11}$ | 17.15 |

Across all datasets, TEL head introduces only a small increase in latency, while keeping the parameter count and FLOPs nearly identical to the original model. The TEL full variant incurs higher inference cost, as expected from its iterative refinement, but remains within a practical range even for large architectures such as ViT-L and GPT-2. These results demonstrate that TEL can be incorporated into large models with modest computational overhead, enabling its stability and accuracy benefits at scale.

### H.7 INTERPRETABILITY AND DIAGNOSTIC SIGNALS OF TEL

TEL exposes internal thermodynamic quantities—enthalpy and entropy gradients, temperature schedules, and free-energy trajectories—that are not available in standard MLPs, residual networks, or implicit layers. These signals arise directly from the refinement dynamics in equation 8–equation 9 and correlate strongly with sample difficulty, model uncertainty, and convergence behavior (§3.4). Across all model scales and datasets, the diagnostic patterns below appear consistently and are supported by TEL's theoretical properties: non-expansiveness (Proposition 2.1), frozen-temperature convergence (Proposition C.4), and the two-time-scale tracking guarantees of Proposition C.6.

**Enthalpy–entropy gradient balance.** The ratio $\rho^{(i)}$ quantifies whether a refinement step is dominated by the anchor term ($\rho < 1$) or by entropy-driven exploration ($\rho > 1$). Across synthetic reconstruction datasets, the mean $\rho^{(i)}$ trajectories exhibit a clear difficulty hierarchy: sinusoid 1D (easy) begins well below 1 and stabilizes quickly, spheres 3D (medium) briefly enters the entropy-driven regime before returning toward 1, and swissroll 3D (hard) shows a pronounced entropy-driven surge followed by gradual relaxation (see Fig. 10). This behavior reflects the gradient scaling guarantees of Lemma C.1.

**Gradient alignment.** The cosine alignment $\kappa^{(i)}$ measures how closely the enthalpy and entropy gradients agree during refinement. Easy examples maintain strong positive alignment, medium examples show a gradual reduction, and hard examples exhibit a temporary loss of alignment due to

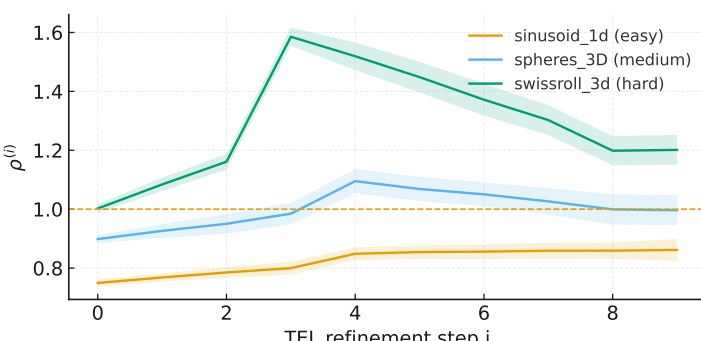

Figure 10: Mean $\rho^{(i)}$ across TEL refinement steps for three synthetic reconstruction datasets. Easy data (sinusoid 1D) remains in the anchor-dominated regime, medium data (spheres 3D) transitions between entropy and anchor regimes, and hard data (swissroll 3D) exhibits a pronounced entropy-driven peak before stabilizing.

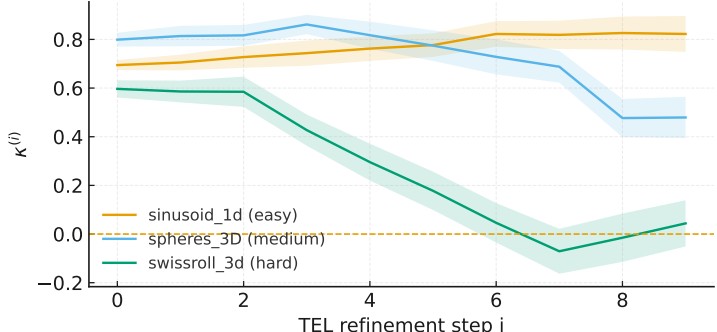

Figure 11: Mean cosine alignment $\kappa^{(i)}$ across refinement steps. Easy examples maintain high agreement, medium examples gradually lose alignment, and hard examples exhibit a temporary decline before stabilizing.

the complex geometry of their underlying manifolds (Fig. 11). These trends illustrate how TEL modulates refinement depending on dataset structure.

**Temperature trajectories.** The adaptive temperature schedule $T^{(i)}$ provides a direct indicator of sample difficulty. All datasets begin at a shared initial temperature, after which easy examples remain low and saturate quickly, medium examples rise more noticeably, and hard examples exhibit the strongest and slowest-saturating temperature increases (Fig. 12). This matches TEL's role in allocating exploratory capacity to ambiguous or complex samples.

**Free-energy descent.** The free-energy $G^{(i)}$ decreases smoothly under stable refinement, with plateaus marking saturation. Easy examples converge rapidly and achieve the lowest plateau, medium examples descend more gradually, and hard examples converge the slowest and stabilize at the highest energy levels (Fig. 13). These behaviors support TEL's non-expansive refinement dynamics across models.

**Cross-scale invariance and practical use.** Across Stages I–III and all architectures, these diagnostics exhibit remarkably consistent structure: (i) $\rho > 1$ marks difficult samples, (ii) negative $\kappa$ indicates conflicting nonlinear corrections, (iii) $\overline{T}$ tracks uncertainty and difficulty, and (iv) $\Delta G$ decreases smoothly under stable refinement. This invariance follows from the shared refinement rule equation 8 and the bounded-gain dynamics in equation 13.

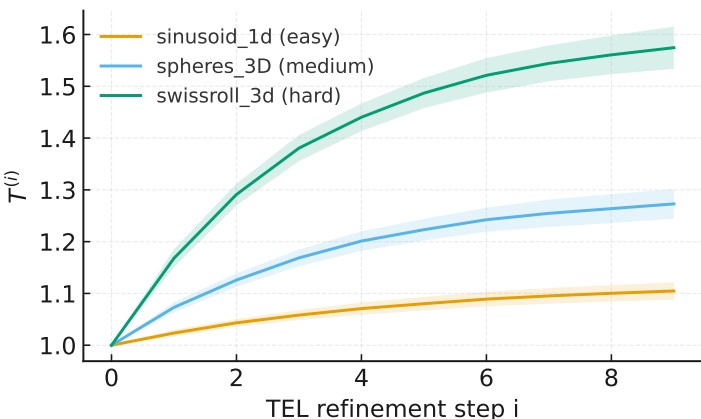

Figure 12: Mean temperature trajectories $T^{(i)}$ across refinement steps. Easy examples remain near the initial temperature, medium examples show a moderate rise, and hard examples exhibit the strongest increase before saturation.

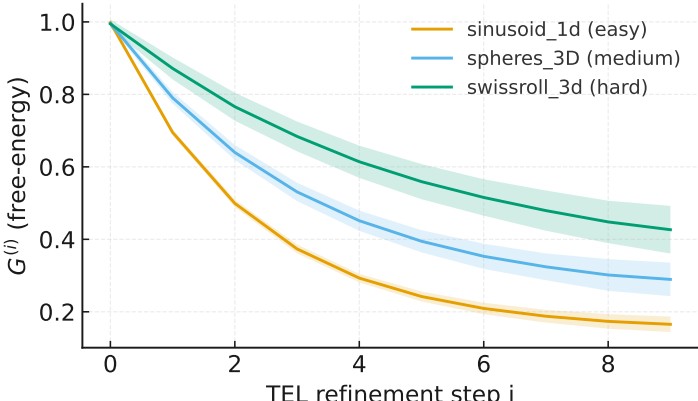

Figure 13: Free-energy evolution $G^{(i)}$ across refinement steps. Easy examples converge fastest and lowest, medium examples stabilize later, and hard examples converge slowest and plateau highest.

Practically, these signals provide lightweight tools for early exits, OOD detection, mislabel identification, and calibration improvements, requiring no architectural changes or auxiliary training objectives.

### H.8 TEL VS. RNNS AND ADAPTIVE-RESIDUAL BASELINE

To enable a fair comparison, we instantiated TEL, the unrolled RNNs, and the adaptive-residual baselines under identical settings: all models use the same 256-dimensional hidden width, the same input/output projections, and the same number of refinement steps $K$. Each method therefore receives exactly $K$ corrective updates, ensuring that differences in behavior arise solely from their update rules rather than disparities in model size or depth. The resulting parameter counts, FLOPs, and performance metrics for all three methods are summarized in Table 7.

The adaptive-residual baseline is a single residual block with a fixed linear "anchor" projection and a Dynamic-ReLU–style gain. A sigmoid gate predicted from the input scales a tanh update on the anchored hidden, producing $h = \text{anchor}(x) + \sigma(g(x)) \odot u(h)$. Parameters are shared across the $K$ passes, but each pass is just another gated residual correction with no coupling to a global objective.

Table 7: TEL vs. 5-step RNN vs. adaptive residual baselines across classification, regression, and reconstruction tasks.

| Dataset | Task | Model | Metric $\pm$ Std | Params | FLOPs |
|---|---|---|---|---|---|
| CIFAR-10 | Classification | TEL (K = 5) | $0.5325 \pm 0.0019$ | $7.90 \times 10^5$ | $1.58 \times 10^6$ |
| | | 5-step RNN | $0.4203 \pm 0.0048$ | $8.55 \times 10^5$ | $8.52 \times 10^6$ |
| | | Adaptive residual | $0.5134 \pm 0.0101$ | $1.64 \times 10^6$ | $3.28 \times 10^6$ |
| California Housing | Regression | TEL (K = 5) | $0.5284 \pm 0.0064$ | $6.86 \times 10^4$ | $4.15 \times 10^4$ |
| | | 5-step RNN | $0.6816 \pm 0.00$ | $1.34 \times 10^5$ | $6.76 \times 10^5$ |
| | | Adaptive residual | $1.0701 \pm 0.00$ | $1.36 \times 10^5$ | $1.40 \times 10^5$ |
| SwissRoll-3D | Reconstruction | TEL (K = 5) | $1.932 \pm 0.0081$ | $6.79 \times 10^4$ | $1.48 \times 10^5$ |
| | | 5-step RNN | $2.032 \pm 0.024$ | $1.33 \times 10^5$ | $7.96 \times 10^5$ |
| | | Adaptive residual | $2.126 \pm 0.013$ | $1.34 \times 10^5$ | $2.67 \times 10^5$ |

The unrolled RNN baseline uses a GRU-like update implemented as a single affine transform on the concatenated input and hidden state, followed by a tanh. The same cell (weights shared across steps) is applied exactly $K$ times and is followed by a single output head. This provides $K$ state re-projections with learned hidden-to-hidden mixing but no anchor or energy constraint.

Although TEL can be written as a sequence of residual-style updates, it is not equivalent to Dynamic ReLU, gated activations, or any learnable-activation mechanism, nor to an unrolled RNN. Learnable activations such as Dynamic ReLU, ACON, or gated MLPs operate by modulating the shape of a static pointwise nonlinearity, typically by predicting slopes, offsets, or mixing coefficients from the input. Their effect is instantaneous: a single forward pass applies the gated activation once, with no notion of refinement, anchoring, or iterative consistency across steps. TEL, in contrast, is built around a multi-step equilibrium refinement in which the representation is progressively corrected relative to a fixed linear anchor. These corrections are not arbitrary or independently learned residual mappings: they are constrained updates derived from a single underlying free-energy objective, which forces each iteration to remain consistent with the same energy–entropy geometry rather than drifting through unrelated nonlinear transformations. Standard learnable activations do not impose any global coherence across steps and therefore cannot ensure that the update sequence follows a descent direction or stays within a stability range.

Moreover, TEL's temperature is not a simple gate applied to an activation. It is a dual variable that governs the balance between structure-seeking (anchor-following) and complexity-seeking (entropy-driven) behavior. Its update depends on global statistics of the intermediate activations, rather than local self-gating heuristics, and its range is explicitly constrained to maintain non-expansive and predictable updates. Dynamic ReLU and similar mechanisms lack this two-timescale structure, lack any coupling between activation geometry and stability, and cannot produce the interpretable diagnostics that TEL naturally yields.

The "$K$-step RNN" baselines represent a different contrast. They repeatedly apply a GRU-style cell with its own learned gates and hidden-to-hidden projections, effectively reprojecting and mixing the hidden state at every iteration. Even when parameters are shared across steps, the recurrence is structurally unconstrained: it has no fixed anchor, no energy-based consistency, and no stability conditions linking one step to the next. The hidden state can drift, rotate, or amplify freely because each update is a general learned transformation rather than a controlled correction. This makes the RNN strictly more parametric and expressive, but also less stable, less interpretable, and fundamentally different from TEL's refinement semantics.

In summary, although all methods are matched in width, compute, and number of update steps, they implement fundamentally different computational principles. TEL performs anchored, energy-consistent refinement governed by an adaptive dual variable; unrolled RNNs repeatedly transform the state using parametric recurrent projections; and dynamic or gated activations apply step-local modulation without any unifying global objective. These structural distinctions, not differences in model scale or training protocol, explain the consistent empirical advantages of TEL over both unrolled-RNN and adaptive-residual baselines.

## H.9 TEL DROP-IN PLACEMENT STUDY

Table 8: TEL drop-in placement study on USPS using LeNet-5. TEL@1 and TEL@2 replace the two hidden FC layers; TEL@3 replaces the classifier head.

| Model | Test Acc. (%) |
|---|---|
| Vanilla | $93.6 \pm 0.012$ |
| TEL@1 | $\mathbf{95.8} \pm 0.014$ |
| TEL@2 | $94.3 \pm 0.017$ |
| TEL@3 | $93.7 \pm 0.011$ |
| TEL-full | $93.9 \pm 0.018$ |

Table 9: CIFAR-10 entropy-estimator study using a shallow TEL block with SiLU activation.

| Estimator type | Extra params | Test acc. (%) | Throughput |
|---|---|---|---|
| Gaussian | $+0$ | $53.25 \pm 0.19$ | $1.16 \times 10^6$ |
| Laplacian | $+0$ | $53.19 \pm 0.21$ | $1.02 \times 10^6$ |
| Student-$t$ | $+0$ | $53.21 \pm 0.25$ | $1.10 \times 10^6$ |
| MLP (pooled moments) | $+10$ | $53.74 \pm 0.71$ | $0.98 \times 10^6$ |

Classic LeNet-5 contains two fully connected (FC) hidden layers after the convolutional blocks ($120 \rightarrow 84$), followed by a linear classifier. Accordingly, TEL@1 and TEL@2 correspond to replacing either of these two hidden FC blocks. For completeness, we also include a TEL@3 configuration in which the final classifier head itself is replaced with TEL. Thus, TEL@3 does not represent a third hidden MLP block, but rather a replacement of the original linear classifier with a TEL refinement module.

We evaluate the following configurations:

- **Vanilla:** standard LeNet-5 head (two hidden FC layers + linear classifier).

- **TEL@1:** TEL replaces the first hidden FC block.

- **TEL@2:** TEL replaces the second hidden FC block.

- **TEL@3:** TEL replaces the classifier head.

- **TEL-full:** TEL replaces all three components (both hidden FC blocks and the classifier head).

All models share the same training configuration (AdamW, identical hyperparameters, early stopping), ensuring that observed performance differences arise only from the placement of TEL.

The results show that TEL placement significantly impacts accuracy as given in Table 8. TEL@1 yields the strongest improvement, reflecting the fact that the first FC block has the widest representation and therefore provides TEL with the largest effective parameter budget. Moving TEL deeper (TEL@2) reduces this width and narrows the space available for refinement, leading to a smaller accuracy gain. When TEL replaces the classifier itself (TEL@3), the representational width collapses to the 10-way output space, dramatically reducing TEL's capacity and producing minimal improvement over the baseline.

Replacing all three components with TEL (TEL-full) does not recover the performance of TEL@1. Although TEL-full applies TEL everywhere, stacking multiple refinement modules introduces excessive refinement complexity, an effect observed consistently across architectures, which ultimately degrades performance.

In summary, TEL is most effective when applied early, where the feature dimensionality is highest. Deeper placement or replacing only the classifier head restricts TEL's capacity, while stacking TEL across all layers amplifies refinement complexity and limits accuracy.

## H.10    ENTROPY ESTIMATOR ARCHITECTURE STUDY

We compare four entropy estimator architectures within a shallow TEL block on CIFAR-10, using SiLU activation, $K=5$ refinement steps, and width 256. Each estimator maps the entropy force $z = \phi_\theta(y)$ to a scalar score $\hat{s}(y)$ used in the TEL refinement update.

**Analytic estimators.** Gaussian, Laplacian, and Student-$t$ estimators all behave similarly in this shallow setting: they require no extra parameters, achieve nearly identical accuracy (around 53%), and differ only slightly in throughput due to minor computational overheads.

**Learned estimator.** A small 2-layer MLP operating on pooled activation statistics offers a modest accuracy improvement (53.74%) but shows the highest variance across runs and is the slowest in throughput due to its additional computation.

Overall, as stated in Table 9 analytic estimators provide comparable performance at minimal cost, while the learned MLP trades speed and stability for a small accuracy gain.

