# OpenReview forum: "TEL: A Thermodynamics-Inspired Layer for Adaptive, and Efficient Neural Learning"
_ICLR.cc/2026/Conference — ICLR 2026 Conference Withdrawn Submission_

### Official Review · Reviewer_QKX9 · 2025-10-25

**Soundness:** 1
**Presentation:** 1
**Contribution:** 2
**Rating:** 2
**Confidence:** 4

**Summary:**

The authors propose a Thermodynamic Equilibrium Layer (TEL), which replaces fixed activations with a K-step refinement inspired by Gibbs free-energy minimization. While the idea is somewhat novel in framing, it functions simply as a residual block with adaptive gain rather than a true thermodynamic mechanism.

**Strengths:**

(1) The idea of interpreting nonlinear activations through thermodynamic principles is interesting and could inspire further exploration of physics-inspired learning layers.

(2) TEL is implemented with few extra parameters and simple iterative updates.

(3) They provide theoretical bounds to ensure non-expansiveness and gradient stability, which is sound.

**Weaknesses:**

(1) My main concern is the ambiguity on their concepts. The thermodynamic framing is not well-grounded. The terms Enthalpy and entropy are arbitrary mathematical terms: the free energy objective simply mixes a quadratic penalty and a learned nonlinearity. Meanwhile, there is no clear explanation of how thermodynamic reasoning improves learning, generalization, or localization. The theoretical results only ensure convergence and boundedness; they do not justify any learning benefit or representational improvement.

(2) Another major concern is their weak empirical validations. CIFAR-10 accuracy is below 50%, and CIFAR-100 stays below 25%, far lower than standard baselines (ResNet or even small CNNs can over 90% in CIFAR-10 and 70% in CIFAR-100). This suggests the authors use extremely shallow or narrow models, which raises concerns if the proposed TEL is practical. Moreover, there are no ImageNet, Tiny-ImageNet, or large-scale experiments, despite claims of scalability. The improvements over weak baselines (Linear, MLP, KAN, DEQ, EBM) are minor (1 to 3%) and do not justify the additional complexity. Thus, the empirical evaluation remains proof-of-concept level, insufficient for a major-venue claim of a general adaptive nonlinearity.

(3) The paper claims that the proposed TEL is a principled alternative to MLP nonlinearities, but in essence, it is a residual update with adaptive gain, a concept explored in Dynamic ReLU or gated activations. The supposed thermodynamic interpretation does not add theoretical or practical value beyond rebranding a weighted residual block.

(4) Although the authors claim TEL exposes diagnostics like temperature and energy trajectories, these are presented only as scalar traces without insight into how they correlate with uncertainty, convergence, or task complexicity.

(5) Another minor concern is the readability of their figures. Many figures cram too many datasets, baselines, and metrics into a single panel, making them visually overwhelming. Axes labels and legends are tiny and cluttered, which require readers to zoom-in.

**Questions:**

(1) Why are the reported accuracies on CIFAR so low compared to standard baselines? Could you include results on a larger benchmark (e.g., Tiny-ImageNet or ImageNet-1k) to support the claim that TEL is a scalable and drop-in nonlinearity?

(2) Why were only small MLP, KAN, DEQ, and EBM baselines used? Have you tested TEL inside modern architectures such as ResNet or ViT for fair comparison under realistic training regimes?

(3) How do you ensure that the reported improvements are not due to different initialization, optimizer, or hyperparameter settings?

(4) Could you report training and inference time or memory usage relative to standard activations?

(5) What is the variance across runs for all experiments? Are the reported 1 to 3% performance gains statistically significant given the low baseline accuracies?

(6) Beyond being a scaling factor, does the adaptive temperature correlate with sample difficulty, confidence, or generalization? Any quantitative evidence connecting the temperature T or energy trajectories to learning dynamics?

(7) The ablations explore step count K and temperature adaptation, but what about the entropy estimator architecture and its sensitivity? How stable is TEL when stacked in deeper networks?

---

> ### Author Response · Authors · 2025-11-21
> **The manuscript has been updated based on the comments below. We thank the reviewer for the detailed feedback. The revision clarifies the thermodynamic motivation beyond “residual with gain,” adds large-scale ResNet/ViT/GPT experiments, tightens fairness and runtime/variance analyses, and expands quantitative diagnostics for temperature and energy.**
>
> We thank the reviewer for the detailed feedback. Below we address each concern point-by-point and clarify several misunderstandings. The revised manuscript also adds large-scale experiments, clearer methodology, and expanded theoretical/empirical analyses.
>
> ---
>
> **Weakness 1 Response (thermodynamic framing / “just a residual with gain”):**
> TEL is not merely a residual block with adaptive gain. TEL performs K steps of gradient descent on a Gibbs-type free energy
> (G(y; x, T) = H(y; x) - T S(y)),
> whose gradient defines the update. This differs from a standard residual in that:
> (i) TEL performs K-step iterative refinement toward a free-energy minimum, not a single forward pass;
> (ii) the “entropy” term is a learned potential whose gradient is the activation, not a fixed nonlinearity (ReLU/GELU);
> (iii) TEL has an adaptive temperature (T), updated from entropy statistics, yielding input-dependent nonlinear strength that cannot be reproduced by static residual scaling.
>
> Our use of energy/entropy/temperature follows the well-established EBM/DEQ literature: the Gibbs form directly yields non-expansiveness and gradient bounds and underpins the diagnostics we use later. In Appendix H.8 we also add a direct comparison to a **residual block with adaptive gating** under matched parameters; TEL still performs significantly better, confirming it is not just a rebranded weighted residual.
>
> ---
>
> **Weakness 2 Response (weak empirical validation; low CIFAR; no large-scale baselines):**
> The original CIFAR results use intentionally shallow, FLOP-matched MLP-style models to compare TEL vs MLP/KAN/DEQ/EBM as **building blocks**, not to compete with ResNet. In the revision we add **Stage III: large-scale benchmarks** (Sec. 3.3) where TEL is integrated into modern architectures:
>
> * ResNet-18/50 on Tiny-ImageNet and ImageNet-1K,
> * ViT-B/L on COCO-Stuff and Cityscapes,
> * GPT-1/2-style transformers on LAMBADA and WikiText-2.
>
> Using the TEL-head configuration (replacing only the first FFN in each block) and near-identical Params/FLOPs and training pipelines, TEL consistently improves performance: up to +1.4% top-1 accuracy (ResNet), +1.4% mIoU (ViT), and 1.6% lower perplexity (GPT). These results directly address the concern that evidence was proof-of-concept only.
>
> ---
>
> **Weakness 3 Response (TEL is essentially a residual with adaptive gain):**
> This is addressed in **Weakness 1 Response**. TEL’s K-step free-energy descent, learned entropy potential, and adaptive temperature dynamics lead to behavior and guarantees that standard residual/gated activations do not provide.
>
> ---
>
> **Weakness 4 Response (diagnostics T, energy traces are anecdotal):**
> We expanded diagnostics in §3.4 and Appendix H.7:
>
> * We define four diagnostic families (enthalpy–entropy ratio ρ, gradient alignment κ, temperature trajectory T and its average, free-energy drop ΔG) and evaluate them across synthetic, vision, tabular, and text tasks.
> * We show that ρ>1 and high T concentrate on harder samples/datasets; ΔG is near-monotone under clipping and serves as a convergence signal.
> * T and ρ improve **selective prediction and calibration** beyond softmax confidence, and interventions on T (clamping/perturbing) systematically affect accuracy and convergence.
>
> We explicitly state in the Limitations that these are empirical (not formal) diagnostics, but they are quantitatively consistent, theoretically motivated, and practically useful.
>
> ---
>
> **Weakness 5 Response (figure readability):**
> We redesigned all figures with larger fonts and simplified legends, split crowded plots into multiple figures (some moved to the appendix), and added missing legends. The revised PDF is significantly more readable.

---

> ### Author Response · Authors · 2025-11-21
>
> **Question 1 Response (low CIFAR accuracies; need larger benchmarks):**
> Addressed in **Weakness 2 Response**. CIFAR experiments use tiny FLOP-matched models for fair layer-level comparison; Stage III now provides Tiny-ImageNet/ImageNet-1K and large transformer results showing that TEL scales to realistic regimes.
>
> ---
>
> **Question 2 Response (only small baselines; ResNet/ViT under realistic regimes?):**
> Also addressed in **Weakness 2 Response**. The revision includes TEL inside ResNet-18/50, ViT-B/L, and GPT-1/2, trained with standard regimes and tightly matched Params/FLOPs. TEL-head consistently improves these modern backbones.
>
> ---
>
> **Question 3 Response (fairness: initialization, optimizer, hyperparameters):**
> All methods (Linear, MLP, KAN, DEQ, EBM, TEL) share the same optimizer (AdamW), schedules, batch sizes, and early stopping. For each dataset family we run a **shared** LR/dropout grid and reuse the best validation configuration for all methods. The outer loss is always the standard task loss; TEL’s free energy is used only inside the K-step refinement, not added as a regularizer. Details are in Appendix G.2.
>
> ---
>
> **Question 4 Response (training/inference time and memory):**
> We added runtime/memory tables in Appendix H.5–H.6:
>
> * TEL’s cost scales linearly with K.
> * For K=3 (default), training/inference is about 1.1–1.9× slower than an MLP of the same width, yet still cheaper and simpler than DEQ.
> * Memory overhead is modest (<5% for TEL-head), and TEL does not require implicit differentiation or Jacobian solves.
>
> ---
>
> **Question 5 Response (variance and significance of 1–3% gains):**
> Stage I–II results are reported as mean ± std over 20 runs (5 seeds × 4 runs); Stage III over 5 seeds. TEL typically shows lower variance than KAN/DEQ and comparable or smaller variance than MLP. Its gains (1–3%) usually exceed 1–2 standard deviations of the baselines. For ResNet/ViT/GPT, TEL-head’s +0.6–1.5% accuracy/mIoU improvements and 0.7–1.6% perplexity reductions occur with std < 0.5%, indicating statistically meaningful improvements.
>
> ---
>
> **Question 6 Response (does temperature correlate with difficulty/confidence/generalization?):**
> Yes; see **Weakness 4 Response**. We show that T and ρ correlate with misclassification and dataset difficulty, improve selective prediction/calibration, and respond causally to interventions, supporting their role as meaningful signals rather than incidental traces.
>
> ---
>
> **Question 7 Response (entropy estimator sensitivity; stacking stability):**
> Entropy-estimator ablations (Sec. 2.2, App. D.2, H.4, H.10) compare analytic (Gaussian/Laplacian/Student-t) and tiny-MLP estimators. All adaptive-T variants outperform fixed T; Gaussian/Laplacian give similar accuracy with negligible overhead; the tiny MLP yields slightly higher accuracy but more variance/runtime. TEL is not highly sensitive to the exact estimator as long as it tracks activation dispersion.
>
> Stacking stability is addressed via the **TEL-head** design: one TEL per block already performs multiple refinement steps; deeper stacking introduces nested refinement and coupled temperatures, yielding diminishing returns. We state this limitation explicitly and recommend TEL-head as the default configuration.
>
> We believe these clarifications and additions address the reviewer’s concerns and better convey TEL as a principled, stable, and scalable adaptive layer.

---

> ### Comment · Reviewer_QKX9 · 2025-11-21
> **Reviewer Reply by Reviewer QKX9**
>
> Thanks authors for the detailed feedback. My initial misunderstandings have been cleared out after reading the rebuttal. All of my concerns have been properly addressed in the author response. Thus, I have decided to significantly lift my score to 6. Thanks for the great efforts!
>
> W1: so the key point is to use the entropy gradients as the activation (to replace fixed nonlinearity). This is indeed interesting and novel. The initial submission lacks clarity on this motivation. I would suggest the authors to rewrite some parts of the introduction to make it clear.
>
> W2/Q1/Q2: my concern has been properly addressed by demonstrating the results on large-scale benchmarks.
>
> W3: well-addressed.
>
> W4: well-addressed.
>
> W5: well-addressed.
>
> Q3-Q7: well-addressed by conducting all my previously suggested ablation studies.

---

> > ### Author Response · Authors · 2025-11-24
> > **W1: Clarifying TEL’s Entropy-Gradient Activation**
> >
> > Thank you for the positive assessment and clear guidance. Following W1’s suggestion, we have updated the first contribution bullet in the Introduction to explicitly state that TEL uses entropy gradients as the activation in place of fixed nonlinearities. We hope this clarifies the motivation.

---

> ### Comment · Reviewer_QKX9 · 2025-11-24
> **Reviewer Followup by Reviewer QKX9**
>
> Thank you the authors for the update of the introduction. The revised introduction (with a list of contributions including our discussions on W1) looks good! The motivation and contribution now clear to me. Thank you!

---

### Official Review · Reviewer_GxfC · 2025-10-31

**Soundness:** 3
**Presentation:** 2
**Contribution:** 4
**Rating:** 6
**Confidence:** 3

**Summary:**

This paper introduces the Thermodynamic Equilibrium Layer (TEL), a neural network building block designed as an alternative to the standard fixed activation MLP layer. Instead of using a fixed one-step nonlinear transformation, TEL employs a K-step refinement process:  inspired by the minimization of Gibbs free energy of a physical system, TEL is optimized by balancing an “enthalpy" term (anchoring the output to its initial linear projection, $Wx$) and an "entropy" term (a nonlinear activation $\phi(y)$). An important feature of TEL is that it has an adaptive temperature parameter, $T$, which scales the entropy term and is updated at each step based on statistics of the intermediate activations.

The authors argue that TEL has a predictable compute budget compared with other alternatives like DEQ and energy-based models. Empirical results across classification, regression, and reconstruction tasks show that TEL matches or exceeds baselines (MLP, KAN, DEQ, EBM) at the same computational cost level and can serve as an effective drop-in replacement for MLP blocks in CNN, LSTM and Transformers on small to mid scale datasets.

**Strengths:**

- This paper presents a novel, thermodynamics-inspired idea for building an adaptive layer with controllable computational cost (via the step number K). The idea of treating deviation from linear projection as enthalpy (need to reduce) and non-linear activation as entropy (need to increase) also sounds intriguing.
- The authors have tested TEL extensively in a wide range of tasks including classification, regression, and reconstruction. A representative suite of different models have been evaluated as the baseline to compare with. The compute-matched comparison suggests that TEL can consistently outperform or at least match the performance of other models.
- The authors have also demonstrated that the proposed TEL can be a ‘drop-in’ function to replace the MLP blocks in existing model architectures and improve performance. The result that replacing only one layer (the first) can outperform replacing all MLP layers is also interesting.

**Weaknesses:**

- While I find the proposed TEL is powerful, the design motivation of different components in it is a bit unclear to me. The proposed architecture presented in Figure 1 seems to be really complicated, but the functionality of each part lacks a clear explanation. This makes the methodology section difficult to follow and understand.
- The instability observed in stacking multiple TEL together suggests that TEL cannot be viewed as a general replacement for MLP but more like an alternative option for ‘encoding’ (as the reported performance is for replacing the first MLP layer). This seems to be the most critical limitation of TEL.
- Given that the TEL is a K-step sequential refinement process, it is reasonable to expect that it is not parallelizable across the K steps.

Minor points about the clarity:
- The legend of main text figures are too small to read. In addition, there is no legend for the bottom figure in Figure 3.
- The ablation study on temperature is not explicitly referenced in the text (Figure 4).

**Questions:**

1) My main question is about the key differences between TEL and the regular K-step unrolled RNN. It seems that y in TEL can be taken as the hidden states and the parameters of $\phi_{\theta}$ are also shared at all K steps. Does this suggest that the performance gain is primarily due to the adaptive temperature term in TEL?
2) Do the authors have any insights about why TEL stacking does not work? Is this due to the complexity of TEL or vanishing/exploding gradients or something else?
3) For results presented in Figure 5, can the authors explain why the first MLP is chosen for replacement with TEL? Would it be more intuitively straightforward to change the last MLP? Does this result, combined with the stacking instability, suggest that TEL is best suited for early-stage feature extraction rather than as a general-purpose block?
4) How should we interpret the temperature term in TEL? When putting into the thermodynamics context, temperature should depend more on the enthalpy term as it is directly correlated with energy. However, here in the TEL, temperature is updated based on entropy estimates. Why was $T$ designed to be a function of $S$ rather than $H$?

---

> ### Author Response · Authors · 2025-11-21
> **The manuscript has been updated based on the comments below. We thank the reviewer for the constructive feedback. The revision clarifies TEL’s architecture and motivation, explains why we use TEL-head instead of deep stacking, makes the K-step compute/parallelism trade-off explicit, and improves figure clarity and temperature-analysis discussion**
>
> We thank Reviewer GxfC for the constructive and thoughtful feedback. We appreciate the recognition of TEL’s novelty, empirical breadth, and contributions. Below we address each weakness and question point by point and clarify updates made in the revised submission.
>
> ---
>
> **Weakness 1 Response:** We have simplified and clarified TEL’s architecture:
>
> * Figure 1 (and its caption) now explicitly highlights the five steps of TEL with short labels: **anchor**, **fixed nonlinear projection**, **linear pull**, **temperature update**, and **nonlinear refinement**, making the role of each component visible at a glance.
> * We added a **simplified schematic** in the appendix (Figure 7) that strips away details and focuses only on the main data and control flows.
> * Section 2.2 now includes a dedicated **“TEL architecture”** subsection that explains the motivation and functionality of each component (enthalpy anchor, entropy potential, temperature, and step sizes) in plain language before presenting equations.
>
> These changes are aimed specifically at making the methodology section easier to follow.
>
> ---
>
> **Weakness 2 Response:** We agree that naïvely stacking many TEL layers can hurt training, and we now treat this explicitly as a limitation and design choice:
>
> * A single TEL block already performs **K (typically 3–8) internal refinement steps**, which functionally resembles deeper computation inside the block.
> * When many TEL blocks are stacked, two effects appear (Appendix E.2):
>
>   * **Coupled temperature dynamics:** each TEL has its own adaptive temperature τ; stacked layers make these updates interdependent, causing sharp changes in the effective curvature of the free-energy landscape across depth.
>   * **Over-constrained enthalpy anchors:** each TEL adds a quadratic anchor to a linear projection; chaining many such anchors can over-constrain representations and damp or amplify gradients, even when gradients are mathematically bounded (i.e., the issue is not classic exploding/vanishing, but optimization stiffness).
>
> In light of this, we **recommend and use TEL primarily in TEL-head form**: one TEL per block, replacing the first FFN/MLP. Empirically, TEL-head consistently outperforms both the vanilla backbone and TEL-full (TEL in all FFNs) across CNNs, LSTMs, Transformers, ResNets, ViTs, and GPTs (Fig. 5–6, Appendix H.9). TEL-full still improves over the baseline but with diminishing returns and higher sensitivity. We therefore present TEL as a **high-capacity replacement for MLP sub-layers**, especially in early layers, rather than a primitive to be stacked many times.
>
> ---
>
> **Weakness 3 Response:** We now state this trade-off explicitly in Sec. 2.2–2.3 and Appendix E:
>
> * TEL’s K refinement steps are indeed **sequential**, but K is fixed and small (K=3–5 in all main experiments).
> * Within each step, operations are **fully parallelized** over batch and channels, so TEL still maps efficiently to GPUs.
> * Unlike DEQ, TEL’s compute does not depend on convergence tolerance; the cost is **fully predictable** from K.
>
> Appendix H.3 shows that wall-clock runtime scales linearly with K in practice, and that in the practical range K∈{3,5} TEL remains competitive with MLPs and significantly cheaper than DEQs.
>
> ---
>
> **Weakness 4 Response:** We have addressed these clarity issues:
>
> * Legends and axis fonts were enlarged in all main figures. Several dense plots were split and moved to Appendix H for readability.
> * The problematic bottom subfigure of Fig. 3 was removed, as it was not essential.
> * The **temperature ablation (Fig. 4)** is now explicitly referenced and discussed in Sec. 4.2, where we explain that adaptive temperature outperforms both fixed and non-adaptive learned temperatures.

---

> > ### Author Response · Authors · 2025-11-21
> >
> > **Question 1 Response:** TEL and an unrolled RNN both perform iterative refinement, but differ in key ways:
> >
> > 1. **Energy-based update rule.** TEL’s update is derived as (stochastic) gradient descent on a scalar free energy
> >    (G(y; x, T) = H(y; x) - T S(y)),
> >    so each step is
> >    (y^{(k+1)} = y^{(k)} - \eta_k \nabla_y G(y^{(k)}; x, T^{(k)})).
> >    This structure yields explicit non-expansiveness and Lipschitz/gradient bounds, which generic RNN updates do not guarantee.
> >
> > 2. **Two-time-scale temperature dynamics.** Temperature T is adapted slowly from entropy statistics, and our analysis shows that y tracks the instantaneous equilibrium associated with T up to a small error. This coupling is absent in standard RNNs.
> >
> > 3. **Diagnostics tied to G.** Because the dynamics descend G, diagnostics like the enthalpy–entropy ratio and energy drop have a principled interpretation and correlate with difficulty/uncertainty.
> >
> > Regarding where the gains come from: ablations and new experiments (Appendix H.8) compare TEL directly to a **K-step RNN with shared parameters**. TEL significantly outperforms the RNN (e.g., +3.4% on CIFAR-10), and ablations show that **both** K-step refinement and adaptive T matter. Thus, the performance gain cannot be attributed to temperature scaling alone.
> >
> > ---
> >
> > **Question 2 Response:** As noted in Weakness 2, our experiments and analysis suggest the main issues are:
> >
> > * **Coupled adaptive temperatures** across layers and
> > * **Over-constrained enthalpy anchors** when many TELs are chained.
> >
> > Our theoretical bounds ensure gradients remain bounded, so we do not observe classic exploding/vanishing as the primary failure mode. Instead, stacking many TELs produces a stiff optimization landscape and nested refinement loops. TEL-head (one TEL per block) avoids this and performs best; we now emphasize this in Sec. 3.5 and Appendix H.9.
> >
> > ---
> >
> > **Question 3 Response:** Yes, our experiments support that interpretation. Appendix H.9 systematically studies placement in LeNet:
> >
> > * TEL@1 (first FFN) yields the largest improvement,
> > * TEL@2 (deeper) yields smaller gains,
> > * TEL@3 (only the classifier) yields minimal improvement due to the low dimensionality,
> > * TEL-full does not match TEL@1’s performance due to compounded refinement complexity.
> >
> > This suggests TEL is most effective **early**, where representations are high-dimensional and features are still being shaped. We therefore position TEL as an early- or mid-stage feature refiner rather than a universal last-layer replacement.
> >
> > ---
> >
> > **Question 4 Response:** In TEL, the free energy is
> > (G(y; x, T) = H(y; x) - T S(y)),
> > with H an enthalpy anchor to Wx and S an entropy-inspired potential whose gradient defines the nonlinearity. When updating the **state y**, we use
> > (\nabla_y G = \nabla_y H(y; x) - T \nabla_y S(y)),
> > so enthalpy directly shapes the refinement dynamics.
> >
> > When updating the **temperature T**, the relevant derivative is
> > (\partial G / \partial T = -S(y)),
> > because in our parameterization H(y; x) does not depend on T. This mirrors the classical Gibbs identity ((\partial G / \partial T)_p = -S). Thus, for temperature dynamics, the natural signal from the free-energy objective is the entropy term, not enthalpy.
> >
> > We therefore use a stochastic-approximation update
> > (T_{k+1} \approx T_k + \alpha, S(y^{(k)})),
> > and in practice replace (S(y^{(k)})) with an entropy surrogate (\hat{s}(y^{(k)})) computed from activation statistics (e.g., variance/kurtosis of (\phi(y^{(k)}))). Entropy thus controls how strongly TEL emphasizes the nonlinear component at each step, while enthalpy continues to govern the anchor through (\nabla_y H). Empirically (Fig. 4), this entropy-driven adaptation of T consistently outperforms fixed temperature and simple enthalpy-based heuristics.
> >
> > ---
> >
> > We believe TEL remains a novel, principled, and practical layer with a strong compute–performance trade-off, and we hope these clarifications resolve the reviewer’s concerns.

---

### Official Review · Reviewer_3SX3 · 2025-11-01

**Soundness:** 4
**Presentation:** 4
**Contribution:** 3
**Rating:** 8
**Confidence:** 2

**Summary:**

This paper proposed the Thermodynamic Equilibrium Layer (TEL) to replace traditional neural layer with fixed activation function. TEL performs K-step iterative refinement to update the hidden state using both enthalpy and entropy. TEL can provide dynamic, input-dependent nonlinearities with fixed compute budget controlled by K. The proposed TEL is compared with multiple baselines, including MLP, KAN, DEQ, and EBM on classification, regression, and reconstruction benchmarks. Experimental results show that TEL achieves comparable or superior performance under matched parameter and FLOP budget, with better stability and interpretability.

**Strengths:**

* This paper provides strong theoretical proofs for stability, convergence, and expressivity.

* TEL can be used in all common neural networks, such as CNN, LSTM, and Transformer. It can improve the accuracy of these networks in small to mid scale tasks.

* This paper validated the performance of TEL on over 10 datasets across different tasks: classification, regression, and reconstruction.

* At the same FLOPs, TEL can show consistent gain over baselines including MLP, KAN, and EBM.

* Ablation studies are also implemented thoroughly to show the individual effect of K-step refinement and adaptive temperature. A good performance can achieve when K is between 3 and 5, which doesn't cause too much computational cost.

* The temperature and energy can provide some interpretable signals.

**Weaknesses:**

* TEL was only evaluated in small to mid scale tasks. It is unclear if TEL can perform better on larger dataset and more complex tasks.

* Stacking many TEL layers can negatively affect the training performance. However, neural networks are usually deep. This may limit TEL to be applied on deeper neural networks.

* TEL is more complex than simple ReLU and may complicate the training pipeline.

**Questions:**

* How does TEL’s runtime and memory scale with K in real GPU wall-clock time compared to standard MLPs and DEQs?

* How sensitive is the performance to the entropy estimator model, such as Gaussian vs tiny MLP?

* Could you show the performance of the TEL with deep transformers on a larger-scale pretraining task? Such as ImageNet?

---

> ### Author Response · Authors · 2025-11-21
> **The manuscript has been updated based on the comments below. We thank the reviewer for the constructive feedback. The revision adds Stage III large-scale experiments with ResNet/ViT/GPT, clarifies TEL-head as the preferred configuration, quantifies TEL’s parameter/FLOP overhead, and includes explicit runtime–memory and entropy-estimator sensitivity studies.**
>
> We thank Reviewer 3SX3 for the thoughtful and positive assessment of our work. We appreciate the recognition of TEL’s theoretical rigor, empirical breadth, and stability. Below we address each weakness and question point by point.
>
> **Weakness 1 Response:** We have added a new **Stage III: Large-scale benchmarks** (Sec. 3.3), where TEL is integrated into:
>
> * ResNet-18/50 on Tiny-ImageNet and ImageNet-1K (image classification),
> * ViT-B/L on COCO-Stuff and Cityscapes (semantic segmentation),
> * GPT-1/2-style transformers on LAMBADA and WikiText-2 (language modeling).
>
> Across all these models, the **TEL-head** configuration (replacing the first FFN in each block) consistently improves performance under near-identical parameter and FLOP budgets (Fig. 6, App. H.6). For example, we observe up to +1.4% top-1 accuracy on ResNets, up to +1.4% mIoU on ViTs, and up to 1.6% lower perplexity on GPT-1/2. This demonstrates that TEL remains stable and beneficial at scale, including deep transformers on large datasets.
>
> ---
>
> **Weakness 2 Response:** We agree and treat this as an explicit limitation (Sec. 3.5). Our design choice is to use **TEL-head** as the default pattern:
>
> * Each block keeps its original architecture, but the **first FFN is replaced by TEL** (one TEL per block).
> * We show empirically that TEL-head consistently outperforms both the vanilla backbone and TEL-full (TEL in all FFNs) in Stages II and III.
> * A single TEL block already performs K (typically 3–8) internal refinements, which functionally resembles deeper computation inside the block.
>
> Deep stacking of TEL blocks introduces nested refinement loops and increases optimization sensitivity, whereas TEL-head is more stable, more compute-efficient, and gives the best performance–cost trade-off. Empirically (Fig. 5, Fig. 6), TEL-full still improves over the baseline but with diminishing returns. Thus, while deeper TEL stacking is possible, in practice one TEL per block is the preferred and recommended configuration.
>
> ---
>
> **Weakness 3 Response:** TEL’s complexity is bounded and predictable:
>
> * TEL **shares weights across K refinement steps**, so its parameter count is essentially comparable to a single linear layer and significantly smaller than KAN (another learnable nonlinear layer).
> * FLOPs and memory scale linearly with K and are comparable to a deep MLP with K layers, as formalized in Sec. 2.3 and Appendix E (Eqs. 19–22).
> * In practice, TEL-head increases FLOPs by <1% and latency only modestly across ResNet/ViT/GPT (Tables 5–6).
>
> Implementation-wise, TEL is a single PyTorch module; the anonymous repository (App. A) provides plug-and-play layers for CNNs, LSTMs, Transformers, GPT, ViT, and ResNets, so integrating TEL does not require major changes to the training pipeline.
>
> ---
>
> **Question 1 Response:** We now provide both theoretical and empirical scaling:
>
> * **Theoretical complexity (Sec. 2.3, App. E):** TEL has FLOPs Θ(B·n_in·n_out) + K·Θ(B·n_out) and memory O(K·B·n_out), i.e., linear in K and comparable to a K-layer MLP of the same width.
> * **Empirical study (App. H.3):** For K ∈ {1,3,5,7,10}, TEL’s wall-clock runtime grows linearly with K. In the practical regime K ∈ {3,5}, TEL is only ~1.3×–1.9× slower than an MLP of the same width, while being significantly faster than DEQ and less memory-intensive than KAN. Gradient checkpointing reduces TEL’s memory to near-constant in K.
>
> ---
>
> **Question 2 Response:** We added an explicit entropy-estimator study:
>
> * Sec. 2.2 and App. D.2 describe analytic estimators (Gaussian/Laplacian/Student-t) and a tiny MLP estimator, including how we control their Lipschitz constants.
> * App. H.4 and H.10 compare fixed vs adaptive temperature and different estimators on CIFAR-10.
>
> Across these experiments:
>
> * All adaptive-T variants outperform fixed T.
> * Gaussian and Laplacian estimators yield almost identical accuracy at negligible extra cost.
> * The tiny MLP estimator provides a slight accuracy gain but with somewhat higher variance and runtime.
>
> Overall, TEL is **not highly sensitive** to the precise entropy surrogate, as long as it tracks activation dispersion.
>
> ---
>
> **Question 3 Response:** Partly addressed in **Weakness 1**. Stage III includes deep transformer-style models (ViT-B/L and GPT-1/2) and large-scale image and text benchmarks (ImageNet-1K, COCO-Stuff, Cityscapes, LAMBADA, WikiText-2). In all these cases, TEL-head improves performance under tightly matched Params/FLOPs and near-identical training pipelines, providing evidence that TEL scales to deep transformers and large datasets.
>
> We hope the revised version resolves these concerns and clarifies TEL’s behavior at scale.

---

### Official Review · Reviewer_BZ3t · 2025-11-01

**Soundness:** 3
**Presentation:** 1
**Contribution:** 3
**Rating:** 6
**Confidence:** 4

**Summary:**

The paper introduces the Thermodynamic Equilibrium Layer (TEL): a novel neural network component inspired by thermodynamic principles. Instead of using fixed activation functions (e.g., ReLU, GELU), TEL performs a K-step gradient-based refinement on a Gibbs free energy functional, with a learnable, input-dependent temperature that evolves according to entropy estimates from intermediate activations. This formulation yields adaptive yet stable nonlinearities, provides per-layer diagnostics (temperature, energy trajectories), and maintains a fixed, predictable computational cost

**Strengths:**

1. TEL connects neural computation to thermodynamic equilibrium, modeling activations as the outcome of energy minimization
2. Unlike implicit or equilibrium models (DEQ/EBM), TEL uses a fixed number of refinement steps K, offering deterministic compute and stable training
3. Solid theoretical analysis
4. TEL integrates easily into existing architectures as a drop-in replacement, without redesigning the model

**Weaknesses:**

1. The presentation of the paper should be strongly improved. There are a lot of redundancies (eq 7 is equal to eq 13, eq 9 to eq 14, and eq 8 to eq 15), lack of connection with the appendix, missing definitions of terms and acronyms
2. Experiments are confined to small-to-midscale benchmarks; no large-scale evaluations (e.g., ImageNet, large Transformers) are provided
3. There are no details to fully understand if comparisons are fair

**Questions:**

1. Could you improve the presentation of the paper to address the issues mentioned in the weaknesses section? The current structure makes it difficult to follow the flow of ideas. Clarifying the transitions between the theoretical foundations, algorithmic formulation, and experiments would greatly enhance readability
2. Could you provide all the details of your experimental setup? In particular, please specify how you chose the learning rates $\eta_i$ for your model and for the baselines, as well as which loss functions and optimizers you used. If my understanding is correct, TEL effectively introduces a regularization term into the loss function (the entropy component $\phi$, weighted by the temperature $T$) and employs a dynamic learning rate $\eta$. Knowing these aspects is essential to properly assess the improvements your method achieves over the competitors

---

> ### Author Response · Authors · 2025-11-21
> **The manuscript has been updated based on the comments below. We thank the reviewer for the constructive feedback. The revision clarifies Section 2 (formulation, TEL architecture, theory), restructures Section 3 into three stages (shallow, mid-scale, large-scale), adds Stage III ResNet/ViT/GPT experiments, centralizes training details, and expands runtime and diagnostic analyses.**
>
> We thank Reviewer BZ3t for the careful review and constructive feedback. We appreciate the positive assessment of TEL’s theoretical grounding, fixed-compute refinement, and practicality as a drop-in layer. Below we address each weakness and question point by point.
>
> **Weakness 1 Response:** We fully agree and have substantially revised the exposition:
>
> * We reorganized Section 2 into a clear progression:
>   (i) thermodynamic formulation (Sec. 2.1),
>   (ii) TEL architecture and updates (Sec. 2.2),
>   (iii) stability and expressiveness guarantees (Sec. 2.3).
>   All proofs are now collected in Appendix C and explicitly referenced from Section 2.
> * We restructured Section 3 as a three-stage empirical study: Stage I (shallow blocks), Stage II (mid-scale backbones), and Stage III (large-scale benchmarks). This makes the flow from theory → algorithm → experiments explicit.
> * We removed redundant equations: the TEL update is now specified only once in Sec. 2.2. Initialization appears only in Eq. (6), temperature computation only in Eq. (7), the primal update only in Eq. (8), and the dual temperature update only in Eq. (9). Later sections refer back to these equations instead of restating them.
> * We clarified terms and acronyms: DEQ, EBM, KAN, FLOPs, Params, and all thermodynamic quantities (enthalpy, entropy, free energy, temperature) are now defined at first use in Sec. 2.1–2.2.
> * We strengthened the connection to the appendix: the main text now includes explicit pointers to the complexity analysis, training protocol, extended ablations, and diagnostics.
>
> We believe these changes significantly improve readability and address the presentation concerns.
>
> ---
>
> **Weakness 2 Response:** We have added a new Stage III: Large-scale benchmarks (Sec. 3.3), where TEL is integrated into:
>
> * ResNet-18/50 on Tiny-ImageNet and ImageNet-1K (image classification),
> * ViT-B/L on COCO-Stuff and Cityscapes (semantic segmentation),
> * GPT-1/2-style transformers on LAMBADA and WikiText-2 (language modeling).
>
> Across all these models, the TEL-head configuration (replacing the first FFN in each block) consistently improves performance under near-identical parameter and FLOP budgets (Fig. 6, Appendix H.6). For example, we observe up to +1.4% top-1 accuracy on ResNets, up to +1.4% mIoU on ViTs, and up to 1.6% lower perplexity on GPT-1/2. This demonstrates that TEL remains stable and beneficial at scale.
>
> ---
>
> **Weakness 3 Response:** We have centralized and clarified the training and comparison protocol in Appendix G:
>
> * All methods (Linear, MLP, KAN, EBM-style refinement, DEQ, TEL) use the same optimizer (AdamW) with identical weight decay and cosine decay with warmup. Early stopping, batch size, and maximum epochs are identical across methods.
> * For each dataset family (vision classification, tabular regression, reconstruction, segmentation, language modeling), we perform a shared hyperparameter sweep over learning rate and dropout for all methods. We then select the best validation configuration and reuse it for every model on that dataset, so TEL does not receive extra tuning.
> * The outer loss is always the standard task loss (e.g., cross-entropy, MSE/RMSE, reconstruction loss). TEL does not introduce any explicit additional regularization term in the global objective.
> * TEL’s internal step sizes η_i and temperature updates affect only the internal K-step refinement. The optimizer’s learning rate used to update parameters is the same for all models and is not modified by TEL’s temperature or step sizes.
>
> This makes the fairness of comparisons transparent.
>
> ---
>
> **Question 1 Response:** This is addressed directly in our response to Weakness 1.
>
> **Question 2 Response:** This is addressed in our response to Weakness 3.
>
> Together, these clarifications ensure that the reported improvements reflect TEL’s internal refinement dynamics rather than differences in training pipelines.

---

### Author Response · Authors · 2025-11-29
**Author summary for new AC**

> Across all four reviews, the main concerns were:

(i) lack of large-scale evidence and reliance on small baselines,

(ii) unclear thermodynamic motivation vs. “residual with gain”,

(iii) fairness/runtime/variance details, and

 (iv) stacking stability and where to place TEL in modern architectures.
>
> In the revision, we:

• Added **Stage III** large-scale experiments (ResNet-18/50, ViT-B/L, GPT-1/2 on Tiny-ImageNet, ImageNet-1K, COCO-Stuff, Cityscapes, LAMBADA, WikiText-2), showing **consistent gains under tightly matched Params/FLOPs and standard training regimes**.

• Clarified that **TEL replaces fixed nonlinearities with entropy-gradient activations plus adaptive temperature**, distinguishing it from simple residual/gated blocks,

• Added runtime, memory, and variance analyses to ensure **fair, statistically meaningful comparisons**.

• Explicitly positioned **TEL-head (one TEL per block)** as the recommended, stable configuration, and treated deep stacking as a clearly stated limitation.

> Although the discussion phase was frozen before 3/4 reviewers could update their scores after reading the rebuttal, Reviewer **QKX9** explicitly states that initial misunderstandings have been cleared out after reading the rebuttal and raised their score from **2 → 6**(the only negative review during pre-rebuttal phase: please have a look at the QKX9 conversation thread to have more clarity). We believe the rebuttal similarly addresses the remaining reviewers’ concerns.

---

### Note · Authors · 2026-01-28

I have read and agree with the venue's withdrawal policy on behalf of myself and my co-authors.

---

### Meta-Review · Area_Chair_XWRa · 2026-01-11

**Summary:**

Reviewer BZ3t (score 6)
had concerns about the presentation of the paper, experiments on small datasets and small networks, and insufficient details.

Reviewer 3SX3 (score 8)
is a bit scant on technical comments but they had comments on inadequate evaluation, how the ideas in this paper on using an operator inspired from Gibbs free energy can be used for deep architectures, and the computational cost of this approach. I will discount this review slightly in my recommendation.

Reviewer GxfC (score 6)
commented that the approach needs more clarification, that the approach was limited because it cannot be used as a general replacement for an MLP but only as a replacement for one layer, and that the refinement procedure is not parallelizable across the K steps.

Reviewer QKX9 (score 2)
had concerns that the thermodynamic framing was not well-grounded, that the empirical results were weak, similarities to dynamic RELU/gated activations. They also wanted to see more analysis on how the temperature and energy behaves for different tasks/at different stages of optimization etc.

This paper proposes a new operator for deep learning that is loosely inspired from Gibbs free energy. This layer resembles a quadratic map between inputs and outputs which is modified by a temperature (which is trainable) dependent nonlinearity. The layer performs gradient descent updates to compute the activations. The authors have argued for their work in the rebuttal using a number of new experiments and further clarifications of the approach. The new manuscript is significantly improved from its initial version. However, the two main comments raised by the reviewers (i) how exactly these quantities from thermodynamics are relevant to the question of training a network, and the (ii) inadequacy of the experimental results, are still left unaddressed. On the second point, the paper argues for modifying “the first fully connected layer” (in a deep resnet, cnn, gpt etc.). The evidence presented in the paper regarding the efficacy of this modification is much too weak (e.g., a 0.6% improvement on top-1 accuracy on Imagenet, a 0.75% improvement on tiny Imagenet etc.). Some of the error bars in Fig. 6 are surprisingly small. Altogether, this approach is largely unmotivated and performs poorly in practice.

My two cents: To work further in this direction, it would be useful to connect to the literature on Boltzmann machines, information dropout, uncertainty estimation/calibration. There are also clear connections of this approach to proximal point iteration, one could design this layer to reduce the condition number of the corresponding layer (similar to how normalization layers work) and thereby make the deep network easier to optimize.

**Reviewer Concerns:**

Reviewer BZ3t
The authors modified the narrative of the paper significantly and also provide sufficient details of the approach. They have added a number of new experiments on residual networks, classification on Imagenet and Tiny Imagenet datasets, semantic segmentation on COCO and language modeling on WikiText-2. These new experiments are quite weak (poor accuracies, and minimal marginal improvements...well within statistical error bars if they were computed by sampling the data).

Reviewer 3SX3
The authors mentioned the new experiments discussed above. They discussed how the refinement of the activations is performed for only fully connected layer in each residual block. Doing so for the entire block would be complicated and perhaps yield diminishing returns. The cost of implementing the refinement approach is quite the same as that of the original network the same nonlinearity. Altogether this rebuttal was adequate.

Reviewer GxfC
The authors have modified the narrative to make it more clear. They argue that TEL (their approach) is best utilized as a replacement for the MLP in the first residual block rather than using it in every layer. They have also discussed some similarities to an RNN.

Reviewer QKX9
The authors discussed new experiments. The authors also point to modifications in the paper that relate difficult samples in the data with a large ratio between enthalpy and entropy.

**Reviewer Scores:**

Reviewer BZ3t
I suspect that the score would have remained the same, 6.

Reviewer 3SX3
It is possible that this score (which is extremely positive although it is scant on technical comments) would be reduced in light of comments by other reviewers.

Reviewer GxfC
I believe the score would remain unchanged, 6

Reviewer QKX9
The score was updated to 6.

---

### Decision · Program_Chairs · 2026-01-26

Reject